# Global ubiquitinome profiling identifies NEDD4 as a regulator of Profilin 1 and actin remodelling in neural crest cells

Iman Lohraseb[1,9], Peter McCarthy[1,9], Genevieve Secker[1], Ceilidh Marchant[1], Jianmin Wu[2,3], Naveid Ali[4], Sharad Kumar [1], Roger J. Daly [5], Natasha L. Harvey[1], Hiroshi Kawabe[6,7], Oded Kleifeld [8], Sophie Wiszniak [1] & Quenten Schwarz[1✉]

The ubiquitin ligase NEDD4 promotes neural crest cell (NCC) survival and stem-cell like properties to regulate craniofacial and peripheral nervous system development. However, how ubiquitination and NEDD4 control NCC development remains unknown. Here we combine quantitative analysis of the proteome, transcriptome and ubiquitinome to identify key developmental signalling pathways that are regulated by NEDD4. We report 276 NEDD4 targets in NCCs and show that loss of NEDD4 leads to a pronounced global reduction in specific ubiquitin lysine linkages. We further show that NEDD4 contributes to the regulation of the NCC actin cytoskeleton by controlling ubiquitination and turnover of Profilin 1 to modulate filamentous actin polymerization. Taken together, our data provide insights into how NEDD4-mediated ubiquitination coordinates key regulatory processes during NCC development.

[1] Centre for Cancer Biology, University of South Australia and SA Pathology, GPO Box 2471, Adelaide 5000, Australia. [2] Kinghorn Cancer Centre & Cancer Division, Garvan Institute of Medical Research, Sydney, NSW 2010, Australia. [3] St Vincent's Clinical School, University of New South Wales, Sydney, NSW 2010, Australia. [4] Bone Therapeutics Group, Bone Biology Division, Garvan Institute of Medical Research, Sydney 2010, Australia. [5] Department of Biochemistry and Molecular Biology, School of Biomedical Sciences, Monash University, Victoria 3800, Australia. [6] Department of Molecular Neurobiology, Max Planck Institute for Experimental Medicine, Goettingen 37075, Germany. [7] Department of Pharmacology, Gunma University Graduate School of Medicine, Showa-machi, Maebashi, Gunma 371-8511, Japan. [8] Faculty of Biology, Technion-Israel Institute of Technology, Technion City, Haifa 3200003, Israel. [9] These authors contributed equally: Iman Lohraseb, Peter McCarthy. ✉email: quenten.schwarz@unisa.edu.au

Neural crest cells (NCCs) are a transient embryonic cell type that form essential components of the craniofacial skeleton and peripheral nervous system[1]. These multipotent cells arise from neuroepithelial precursors under instruction of several morphogens from the WNT, Hedgehog, BMP, TGF-β and Notch families that promote expression of core specification transcription factors to control an epithelial to mesenchymal transition. In addition to their roles in establishing bona fide NCCs, these morphogenic signalling pathways and core transcription factors play central roles in NCC survival, NCC stem-cell identity and differentiation into bone, cartilage, peripheral neurons and glia. Although coordination of these signalling pathways is critical for NCC development and differentiation, how they are integrated remains unknown.

Ubiquitination plays essential role in a variety of cellular processes by modulating the levels and/or function of target proteins. These can include degradation by the proteasome or lysosome, modulation of stability, change in subcellular localisation or modification of biochemical properties[2]. Ubiquitin is a highly conserved 8 kDa protein that is covalently coupled to lysine residues on target proteins by the combined functions of activating (E1), conjugating (E2) and ligating (E3) enzymes[3]. E3 ubiquitin ligases perform the final step in the ubiquitination process by catalysing the transfer of ubiquitin from an E2 enzyme to the substrate. As such, E3 ligases are essential for specific substrate ubiquitination, with the ability to target multiple substrates uniquely positioning them to modulate several signalling pathways simultaneously. The ubiquitin moiety contains seven lysine residues that can themselves be ubiquitinated, thereby leading to various ubiquitin chain lengths and topologies on substrate proteins. Substrates can be monoubiquitinated and/or polyubiquitinated on lysine residues to modulate different functional outcomes. However, differences in the way individual moieties in a polyubiquitin chain are coupled can have profound differences in functional outcomes leading to a diverse set of biological roles for ubiquitination. For example, linkages to K48 and K11 on intra-chain ubiquitin units are generally associated with proteasomal degradation. Conversely, K63 linkage is mostly associated with non-proteasomal outcomes including protein trafficking and altering protein function[4]. To date, the role of protein ubiquitination in NCCs is ill-defined and has not been examined at the proteome level[5].

We recently found that protein ubiquitination is involved in NCC development, demonstrating an essential requirement for the E3 ligase NEDD4 in cranial NCCs[6,7]. Mice lacking NEDD4 in NCCs have aberrant NCC death, decreased osteoblast proliferation and reduction in craniofacial bone and peripheral neurons. NEDD4 is the prototypical protein of a family of HECT type E3s that comprise N-terminal C2 domains, 2-4 WW domains and a C-terminal HECT[8,9]. Although many putative NEDD4 targets have been identified using in vitro studies, few have been validated in vivo, and none have been identified in NCCs. A comprehensive understanding of NEDD4 targets in NCCs is therefore required to elucidate how NEDD4 promotes NCC development. Moreover, profiling of the molecular topology of ubiquitination is essential to define how the ubiquitin system modulates NCC development to control formation of the craniofacial skeleton and peripheral nervous system.

Here we have profiled the ubiquitination landscape in NCCs using a quantitative proteomics approach to identify NEDD4 substrates that regulate NCC development. Our results demonstrate that loss of NEDD4 influences the ubiquitination of a large number of NCC proteins and has profound effect on global ubiquitin usage. We find enrichment of candidate NEDD4 substrates in cellular processes critical to NCC development and define a role for NEDD4 as a regulator of the actin cytoskeleton. NEDD4 was found to ubiquitinate the actin-binding protein Profilin 1 (PFN1) to regulate protein turnover and function. Reduction of $Pfn1$ levels in $Nedd4^{-/-}$ NCCs normalized actin polymerisation, therefore uncovering an important regulatory mechanism of actin dynamics essential for craniofacial and peripheral nervous system development.

## Results

**Global proteomics analysis of NCCs.** To uncover ubiquitination sites influenced by NEDD4 in NCCs we used a quantitative mass spectrometry approach combining analyses of cognate protein abundance and affinity enriched ubiquitinated peptides (Fig. 1a). For these experiments, a SILAC-labelled NCC line (NCU10K) was treated for 48 h with previously validated control and Nedd4 siRNAs that robustly knockdown Nedd4[7] and induce changes in cell morphology (Supplementary Fig. 1). Lysates from these two conditions were combined, digested with trypsin, fractionated and the resulting peptides used to quantitate total protein abundance (pre-enrichment) and for ubiquitin remnant motif (K-ε-GG) enrichment (post-enrichment)[10]. K-ε-GG enrichment uses antibody-based purification of diglycine remnant containing peptides that mark ubiquitinated proteins after tryptic digestion. Pre- and post-enrichment samples were analysed by LC-MS/MS from two biological replicates.

**High-throughput expression analyses uncover essential roles for NEDD4 in NCC signalling pathways and regulation of the actin cytoskeleton.** Pre-enriched total protein analysis identified quantitative data for 4381 proteins across both biological replicates using an FDR of 1% (Fig. 1b). Between replicates, we observed a high correlation of H/L ratios (3698 proteins were common to both replicates out of the 4988 observed; squared Spearman correlation co-efficient = 0.7156) (Fig. 1c). Protein abundance from the pre-enriched fractions approximated a normal distribution with no overt effect of Nedd4 knockdown on total protein abundance (log2 H/L mean = −0.07; STDEV = 0.596) (Fig. 1d). This dataset identified 190 proteins that were significantly increased or decreased in abundance due to Nedd4 knockdown (Supplementary Data 1). To validate our global MS data we performed western blot for several proteins that were altered in abundance following Nedd4 knockdown. Consistent with our proteomics data, SOX10 and ZEB2 were decreased, ERBB3 was increased and FNDC3B was expressed normally in Nedd4 knockdown cells (Supplementary Fig. 2).

We next compared total protein abundance to gene expression data obtained by microarray (Supplementary Data 1 and 2). Overall the two data sets showed low correlation (squared Spearman correlation co-efficient = 0.21), which is consistent with other reports comparing high-throughput proteomic and transcriptomic profiling of matched samples[11,12]. Of the 190 differentially expressed proteins, less than a fifth (37) showed significant changes in mRNA expression (Fig. 1e; magenta dots for increased gene expression, green dots for decreased). This suggests that the abundance of a considerable number of proteins is affected by post-transcriptional mechanisms as a result of $Nedd4$ knockdown. Process network analysis on genes that were significantly affected at the mRNA level ($p < 0.05$) by $Nedd4$ knockdown (David, KEGG) identified significant enrichment within signalling pathways that regulate NCC development (Fig. 1f and Supplementary Data 3). This included several intracellular signalling pathways, including RAP1, AKT, MAPK and RAS, the Notch signalling pathway, and regulation of the actin cytoskeleton. Consistent with a lack of correlation between the total protein and microarray analyses, process network analysis was unable to identify any overlap in enriched molecular pathways between the two data sets (Supplementary Data 3 and 4). However, alterations to genes that regulate the actin cytoskeleton is in strong agreement with the cell morphology change induced by $Nedd4$ knockdown[13].

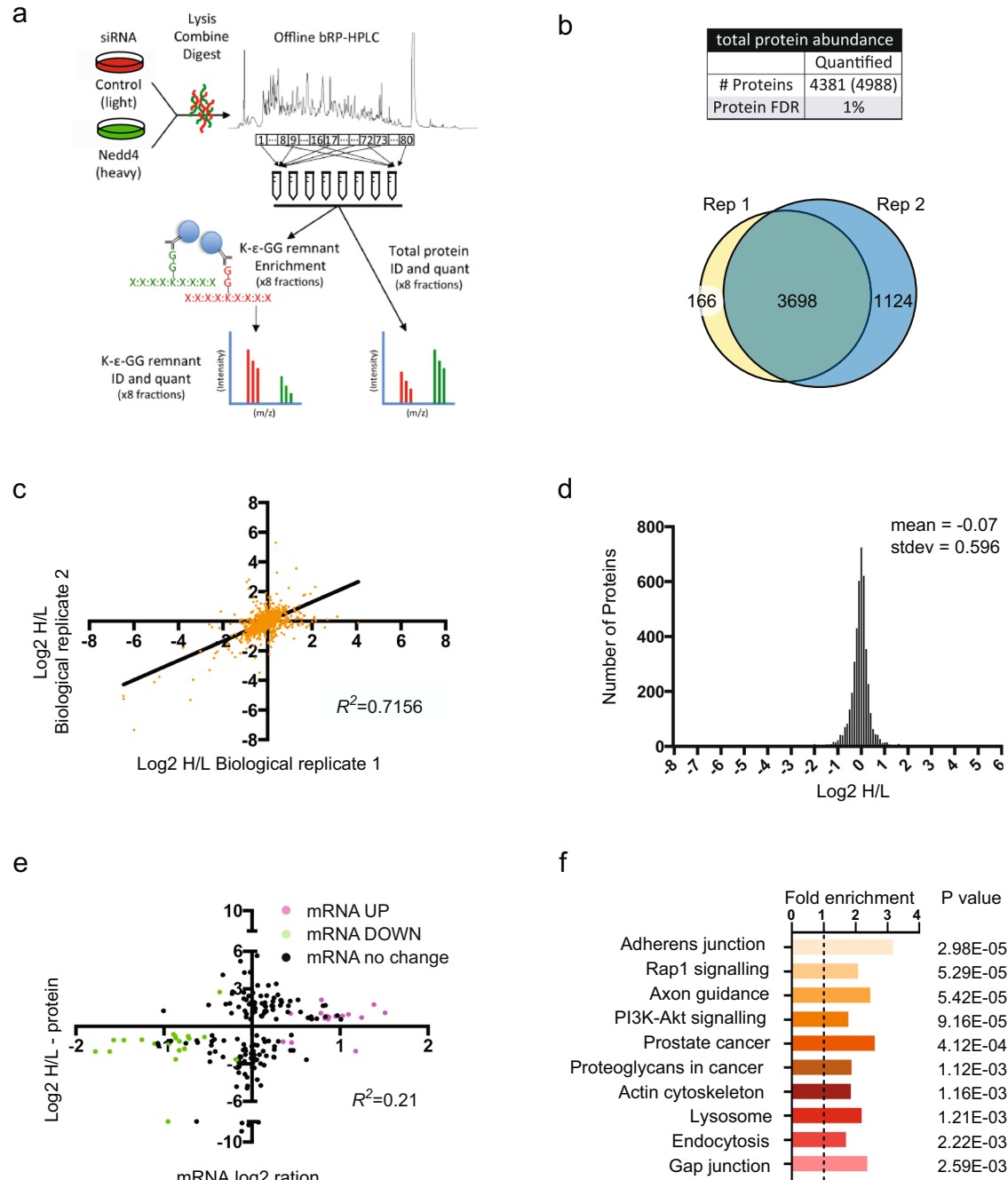

**Fig. 1 Quantitative proteomic profiling for NEDD4 regulated proteins and molecular pathways. a** Workflow for analysis of ubiquitin remnant enriched and pre-enrichment peptides. SILAC labelled NCU10K cells treated with low GC (control, light - red) or *Nedd4* (heavy, green) siRNA were lysed in 8 M urea, equal amounts (5 mg each) were combined and digested with trypsin. The resulting peptides were fractionated by offline basic reversed-phase HPLC into 80 fractions that were then pooled in a discontinuous pooling strategy into 8 fractions. The bulk of each fraction was then subjected to ubiquitin remnant (K-ε-GG remnant) enrichment and enriched peptides identified and quantified by LC-MS/MS. A small amount of each fraction was also analysed prior to enrichment for total protein identification and quantification for use in subsequent normalisation and data analysis. **b** The combined number of quantified protein groups (total number observed in brackets) and Venn diagram showing overlap of protein group IDs from the total protein (unenriched) fractions across 2 biological replicates (rep). **c** Correlation of log2 H/L ratios for each protein group common to both replicates ($R^2 = 0.7156$). **d** Frequency distribution of log2 H/L ratios for protein groups with a mean of $-0.07$ and standard deviation of 0.596. **e** Correlation of total protein abundance (Y-axis) to gene expression data (X-axis) ($R^2 = 0.21$) for proteins with altered abundance. Proteins indicated in green had a statistically significant reduction in gene expression (step-up $p < 0.05$). Proteins indicated in magenta had a statistically significant increase in gene expression (step-up $p < 0.05$) as determined by two-sided ANOVA with adjustments made using Benjamini-Hochberg Step-Up controlling procedure. **f** Top 10 annotations from process network analysis of differentially expressed genes at the mRNA level using DAVID. Fold enrichment and *p*-values derived using Fisher's exact test with adjustments using Bonferroni, Benjamini and FDR methods are shown.

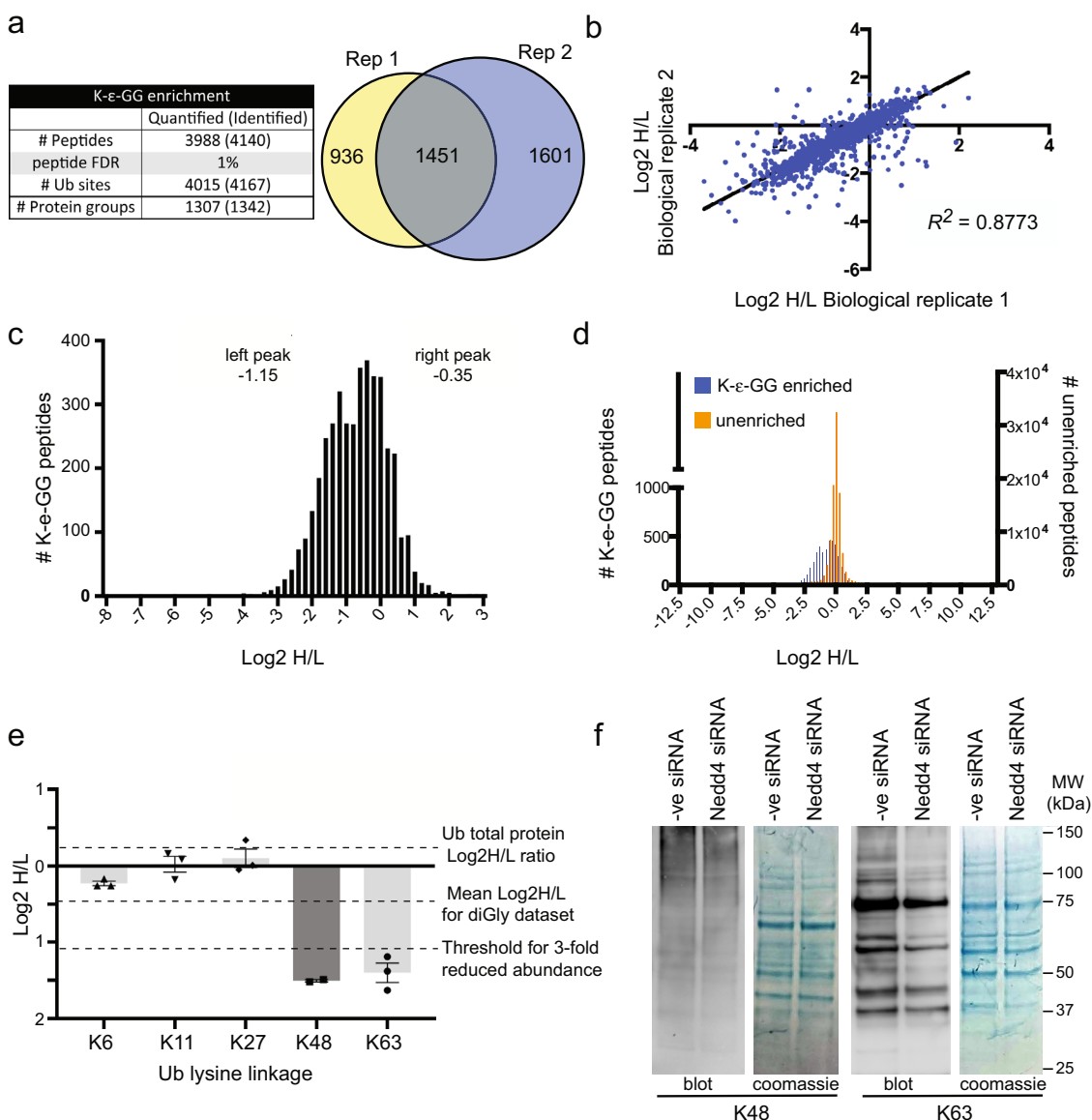

**Fig. 2 Quantitative proteomic profiling of NEDD4 regulated ubiquitination events. a** Number of quantified ubiquitin (ub) remnant peptides and cognate protein groups (total number observed given in brackets). Venn diagram shows overlap of ubiquitin remnant peptides across 2 biological replicates (Rep). **b** Correlation of log2 H/L ratios between replicates ($R^2 = 0.8773$). **c** The frequency distribution of log2 H/L ratios for ubiquitin remnant enriched (K-ε-GG) peptides has a large subset of ubiquitin remnant enriched peptides shifted left. The left peak has a mean of −1.15 and right peak −0.35. **d** Overlay of log2 H/L ratio distributions from ubiquitin remnant enriched (K-ε-GG) peptides (blue) and pre-enrichment peptides (orange) shows ubiquitin remnant peptide abundance is shifted left compared to the pre-enrichment peptide samples. **e** Log2 H/L ratios of ubiquitin remnant peptides derived from inter-ubiquitin isopeptide bonds. There is a specific reduction in abundance of ubiquitin-derived peptides with K48 and K63 ubiquitin linkages following *Nedd4* siRNA knockdown. Total ubiquitin abundance was not altered in the *Nedd4* siRNA treated cells. Quantitation of the ubiquitin linkages was performed from 3 independent experiments. Data are presented as mean values +/− SEM. **f** Reduction of ubiquitin remnants for K48 and K63 linkage was confirmed by western blot using linkage-specific antibodies. Note that equal amounts of lysate loaded as indicated in Coomassie brilliant blue-stained membrane. Blots are representative of 3 repeats. Uncropped images are provided as a Source data file.

**NEDD4 regulates the ubiquitinome of NCCs.** Mass spectrometry of K-ε-GG enriched samples identified 4140 unique ubiquitin remnant containing peptides representing 4167 ubiquitination sites at an FDR of 1% across both biological replicates (Fig. 2a and Supplementary Data 6). Quantitative data were obtained for 3988 peptides corresponding to 4015 unique ubiquitination sites on a total of 1307 proteins. While a large proportion of quantified peptides were identified in only one replicate (2537 unique to 1 replicate vs 1451 common to both), there was strong correlation of H/L ratios observed between

experiments (Fig. 2b) (squared Spearman correlation co-efficient = 0.8773). Consistent with other published studies using ubiquitin remnant enrichment to examine ubiquitination[14] we observed more than one unique ubiquitin remnant carrying peptide for ~54% of proteins identified in our study (Supplementary Fig. 3).

Unexpectedly, *Nedd4* knockdown affected the ubiquitination of a large subset of peptides. Log transformed H/L ratios of ubiquitinated peptides found a bimodal distribution with a subset of peptides showing a leftward shift indicating reduced

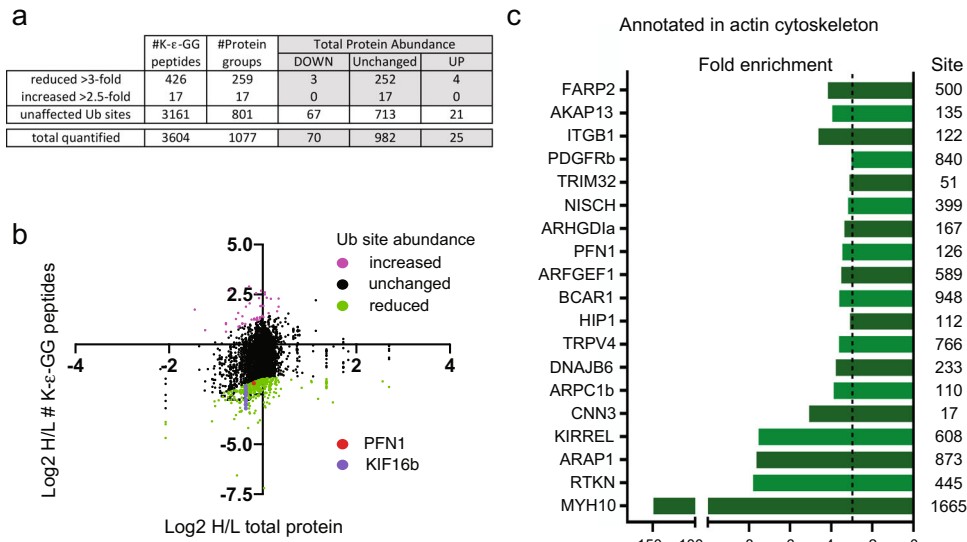

**Fig. 3 Identification of putative NEDD4 targets. a** Tabulated values for the number of ubiquitin (ub) remnant enriched peptides satisfying criteria of being affected by *Nedd4* knockdown, the number of corresponding protein groups and a breakdown of abundance changes for proteins in either group. A breakdown of these values across the entire dataset (for all ubiquitin remnant peptides with cognate proteins in the pre-enrichment dataset) is shown on the last line. **b** 2D plot of log2 H/L ratios for all ubiquitin remnant enriched (K-ε-GG) peptides (y-axis) versus the log2 H/L ratio of their cognate protein (x-axis). Ubiquitin remnant enriched peptides with over 3-fold decrease compared to matched total protein are shown in green. Ubiquitin remnant peptides over 3-fold increase compared to matched total protein are shown in magenta. Ubiquitination events on PFN1 are shown in red, and KIF16B in purple. **c** Proteins with peptides containing over 3-fold reduced ubiquitination that are annotated in the actin cytoskeleton pathway. Only the top-ranked site for each protein is shown with corresponding lysine residue.

abundance in cells treated with the *Nedd4* siRNA (Fig. 2c). This effect was not observed in peptides from the matching pre-enriched samples, which approximated a normal distribution (Fig. 2d). This overall change in abundance was underpinned by a reduction in K63 and K48 ubiquitin linkages following siRNA knockdown of *Nedd4* (Fig. 2e), which were validated by immunoblotting (Fig. 2f). The reduction in specific ubiquitin linkage usage was not coincident with a reduction in overall ubiquitin abundance as determined in our MS data and confirmed by western blot (ubiquitin log2H/L ratio 0.068; Fig. 2e and Supplementary Fig. 4).

**Identification of NEDD4 targets in NCCs**. To identify putative NEDD4 targets we compared the abundance of ubiquitin remnant enriched peptides to the abundance of their cognate protein (pre-enrichment). Of the 3988 ubiquitin remnant enriched peptides for which quantitative data were available, 3604 had matching quantitative data in our pre-enriched data set (Fig. 3a). As *Nedd4* knockdown produced a bimodal distribution of ubiquitinated peptides we only considered ubiquitinated lysine residues with 3-fold decrease (green spots) or 3-fold increase (magenta) relative to their cognate protein to be affected. Using these criteria we identify 426 ubiquitination sites (259 unique proteins) that are reduced in our data set and 17 ubiquitination sites (17 unique proteins) that are increased (Supplementary Data 6). Over 95% of proteins with affected ubiquitination sites following *Nedd4* knockdown had no change in abundance (269 out of 276, Fig. 3b), with only 4 of these proteins having a significant increase. This finding suggests that many of the ubiquitination events influenced by NEDD4 (directly or indirectly) at steady state are not involved in controlling protein turnover. Process network analysis with proteins that had altered ubiquitination identified significant enrichment in signalling pathways that overlapped well with the gene data (Supplementary Data 3 and 5), including RAP1 signalling, AKT signalling and regulation of the actin cytoskeleton. Notably, 19 of the proteins with over

3-fold changes in ubiquitination levels (up or down) but not protein abundance, have established roles in regulating the actin cytoskeleton (Fig. 3c).

Using the protein interaction network analysis platform (PINA)[15] we found that 111 of the proteins with reduced ubiquitination (~40%) have previously been reported to interact with NEDD4 (Supplementary Data 7 and 8). To expand this analysis we immunoprecipitated NEDD4 from LIGHT SILAC labelled NCCs and compared the identified co-interacting proteins to an IgG control IP from HEAVY SILAC NCCs using LC-MS/MS (Supplementary Fig. 5). Of the 276 proteins identified in our ubiquitin screen, 81 were found to co-immunoprecipitate with NEDD4 with over 1.5-fold enrichment compared to IgG control (Supplementary Data 7 and 8). Taken together, these high-throughput MS approaches uncover a comprehensive suite of NEDD4 interactions and putative targets controlling NCC development.

**Validation of NEDD4 targets**. The morphological changes in NCCs treated with *Nedd4* siRNA prompted us to validate NEDD4 targets with established roles in cytoskeletal function. Using a cell-based ubiquitination assay we compared ubiquitination of putative target proteins in the presence of Flag-tagged NEDD4 or a catalytically dead version of NEDD4 (NEDD4-CS) that is unable to transfer ubiquitin to its target proteins[16]. We first tested the kinesin motor protein, KIF16B, that transports early endosomes to the plus end of microtubules[17]. KIF16B had several lysine residues with over 3-fold reduced ubiquitination in *Nedd4* knockdown cells (K633, K679, K781, K815, K846, K858, K863, K869, K902, K932, K960, K1019, K1155, K1191, K1300 and K1312) and contains a 'PY-like' motif that has been suggested to act as a recognition site for NEDD4 interactions[18]. KIF16B was also identified as an interactor in our MS SILAC approach (Supplementary Data 7). Immunoprecipitation of KIF16B with GFP-Trap beads was performed on protein lysates from 293 T cells co-transfected with Flag-tagged Nedd4 (or Flag-tagged

Nedd4-CS), HA-tagged ubiquitin and GFP-tagged Kif16B. Blotting for HA to detect ubiquitinated KIF16B showed mild enrichment of ubiquitin in the presence of NEDD4 compared to NEDD4-CS with a unique ubiquitinated band and increased signal intensity of other bands and a polyubiquitin smear above 150 kDa (Supplementary Fig. 6a). In a similar manner we next explored the actin-binding protein PFN1, that had several lysine residues with over 2-fold reduced ubiquitination (K54, −2.68-fold reduced; K70, −2.89-fold reduced) and one lysine residue with over 3-fold reduced ubiquitination (K128, −3.46-fold reduced). Despite lacking any L/PPXY or 'PY-like' motifs, PFN1 was also identified as a NEDD4 interactor in our MS SILAC approach (Supplementary Data 7). Blotting of immunoprecipitated GFP-PFN1 for HA identified strong enrichment of ubiquitinated PFN1 in the presence of NEDD4 compared to NEDD4-CS with a predominant ubiquitinated band at approximately 42 kDa and an increased polyubiquitin smear above 50 kDa (Fig. 4a). The size increase of 8–10 kDa is consistent with the predominant band representing monoubiquitination of PFN1 (arrow, Fig. 4a). Taken together, these experiments provide strong validation that our MS approach uncovers a large number of NEDD4 targets and identifies NEDD4 as an E3 ligase for KIF16B and PFN1.

**NEDD4 regulates stability of PFN1.** As NEDD4 induces pronounced ubiquitination of PFN1 we investigated the possibility that NEDD4 may regulate PFN1 function. We first validated the interaction between NEDD4 and PFN1 using co-immunoprecipitation of tagged expression constructs in 293T cells, in comparison to control IgG immunoprecipitation (Fig. 4b). To explore the functional roles of NEDD4 we generated $Nedd4^{-/-}$ NCC (NCU10K) and control cell lines using CRISPR genome editing (Fig. 4c). Immunoprecipitation of GFP-PFN1 expressed in these CRISPR edited cell lines demonstrated a mild but significant reduction of protein ubiquitination in the absence of $Nedd4$ (Fig. 4d). Ubiquitination can induce a variety of post-translational outcomes including degradation and functional alterations. Although total levels of PFN1 were not significantly altered in our pre-enrichment MS analysis, western blot of protein lysates from $Nedd4^{-/-}$ NCCs indicated that PFN1 levels are mildly increased in the absence of NEDD4 (Fig. 4c). To explore the possibility that NEDD4 regulates PFN1 turnover we performed a cyclohexamide chase assay in which protein degradation is monitored after inhibition of new protein synthesis. After 9 h of treatment with cycloheximide, PFN1 had degraded by around 40% in control cells, which was blocked by inhibition of the proteasome and lysosome with MG132 and chloroquine (Fig. 4e, f and Supplementary Fig. 6b). Notably, degradation of PFN1 was also blocked in $Nedd4^{-/-}$ NCCs (Fig. 4e, f and Supplementary Fig. 6b), therefore demonstrating that NEDD4 acts to promote PFN1 turnover in this cell type.

**NEDD4 regulates the actin cytoskeleton.** PFN1 regulates actin polymerization by promoting the recruitment ATP-actin monomers to growing filaments. The effect of NEDD4 deficiency and altered PFN1 function on the actin cytoskeleton was therefore examined by staining F-actin with fluorescently conjugated phalloidin. Consistent with previous studies in which PFN1 levels have been manipulated[19], $Nedd4^{-/-}$ NCCs had significantly increased actin filaments, with increased stress-fibres located throughout the cell (Fig. 5a). Quantitation of actin levels and stress-fibre localization by measuring radial mean intensity (RMS) of actin in 5 concentric bins from the centre to the edge[20] identified increased F-actin density throughout $Nedd4^{-/-}$ NCCs (Fig. 5a, b), with average fluorescence intensity of the whole cell also being significantly increased (Fig. 5c). A similar increase in filamentous actin density was observed in $Nedd4$ siRNA treated

NCCs migrating away from trunk neural tube explants grown ex vivo (Fig. 6a). Alterations to the assembly and disassembly of actin filaments as a result of fluctuations to PFN1 levels and function have dramatic effects on cell migration, with lower levels of PFN1 found to promote migratory phenotypes in several metastatic cancers[21]. Consistent with NEDD4 regulating PFN1 function in NCCs, we found that $Nedd4$ siRNA treated NCCs migrated a shorter distance from trunk neural tube explants (Fig. 6b) and that migration of $Nedd4^{-/-}$ NCCs toward FCS was decreased compared to controls (Supplementary Fig. 7). To explore a potential role for NEDD4 in regulating actin polymerization in vivo we next analysed F-actin in $Wnt1-Cre; Nedd4^{fl/fl}; Z/EG$ embryos which lack NEDD4 in all NCCs and their derivatives, and in which NCCs are labelled by GFP. Quantitation of actin levels in GFP + lineage traced trunk NCCs positioned around the dorsal aorta of E9.5 embryos identified a significant increase in F-actin content similar to that of $Nedd4^{-/-}$ NCCs grown in culture (Fig. 6c, d). This increase in actin was coincident with mild NCC migration defects in E9.5 $Wnt1-Cre; Nedd4^{fl/fl}$ embryos. Thus, we found that actively migrating and pioneering SOX10 + trunk NCCs had migrated a shorter distance in somites 9–10 of $Wnt1-Cre; Nedd4^{fl/fl}$ embryos compared to NCCs in the corresponding somites of stage-matched controls (Fig. 6e, f and Supplementary Fig. 8). Importantly, we found no alteration to the number or NCCs migrating in the somites of $Wnt1-Cre; Nedd4^{fl/fl}$ embryos or to the total distance NCCs could migrate (i.e. compare somites 5–10 in which NCCs are actively migrating with somites 11–14 in which NCCs have already arrived at primordia of the sympathetic chain next to the dorsal aorta; Fig. 6e, f), suggesting that this deficiency results from altered migration dynamics rather than structural defects within their environment. We next examined the actin content in cranial NCC derived osteoblasts. Consistent with our previous analysis of craniofacial bone formation, we observed deficiency of RUNX2 + bone precursors in the mandible of E12.5 in $Wnt1-Cre; Nedd4^{fl/fl}$ embryos (Fig. 6g). Moreover, there was a striking increase in F-actin in these cranial NCC derived cells. Taken together, this identifies a key role for NEDD4 in regulating actin polymerisation in NCCs.

**NEDD4 regulates PFN1 function to control actin polymerisation.** To further examine the role of NEDD4 deficiency and altered PFN1 function on actin polymerisation we performed cytochalasin D (Cyto D) washout experiments using $Nedd4^{-/-}$ NCCs. NCCs were initially transfected with Actin-GFP and imaged with spinning disc confocal microscopy to visualize actin polymerisation in real time. Cyto D treatment induced a similar level of cellular contraction and actin depolymerization in $Nedd4^{-/-}$ and control NCCs within 30 min (Fig. 7a). However, following removal of Cyto D, cell spreading and F-actin stress fibre recovery was quicker and denser in $Nedd4^{-/-}$ NCCs, reaching pretreatment levels within 120 min. In contrast, stress fibres were still reduced in control NCCs 180 min post Cyto D washout (Fig. 7a). Increased recovery and abundance of filamentous actin and stress fibres was also evident in fixed $Nedd4^{-/-}$ NCCs labelled with phalloidin (Fig. 7b). Quantitation of actin levels and stress-fibre localization shows that after 180 min of recovery there was significantly more actin in $Nedd4^{-/-}$ NCCs (Fig. 7c). We next tested if the aberrant increase in actin polymerisation resulted from altered PFN1 turnover. To reduce the levels of $Pfn1$ in NCCs we optimized an siRNA approach to knock down total protein by only 30–40% (Fig. 7d). Treatment of $Nedd4^{-/-}$ NCCs with $Pfn1$ siRNA increased the time for actin recovery post Cyto D washout, similar to controls. Moreover, actin levels and localization in $Nedd4^{-/-}$ NCCs treated with $Pfn1$ siRNA were indistinguishable to controls (Fig. 7b, c). As PFN1 promotes the formation of

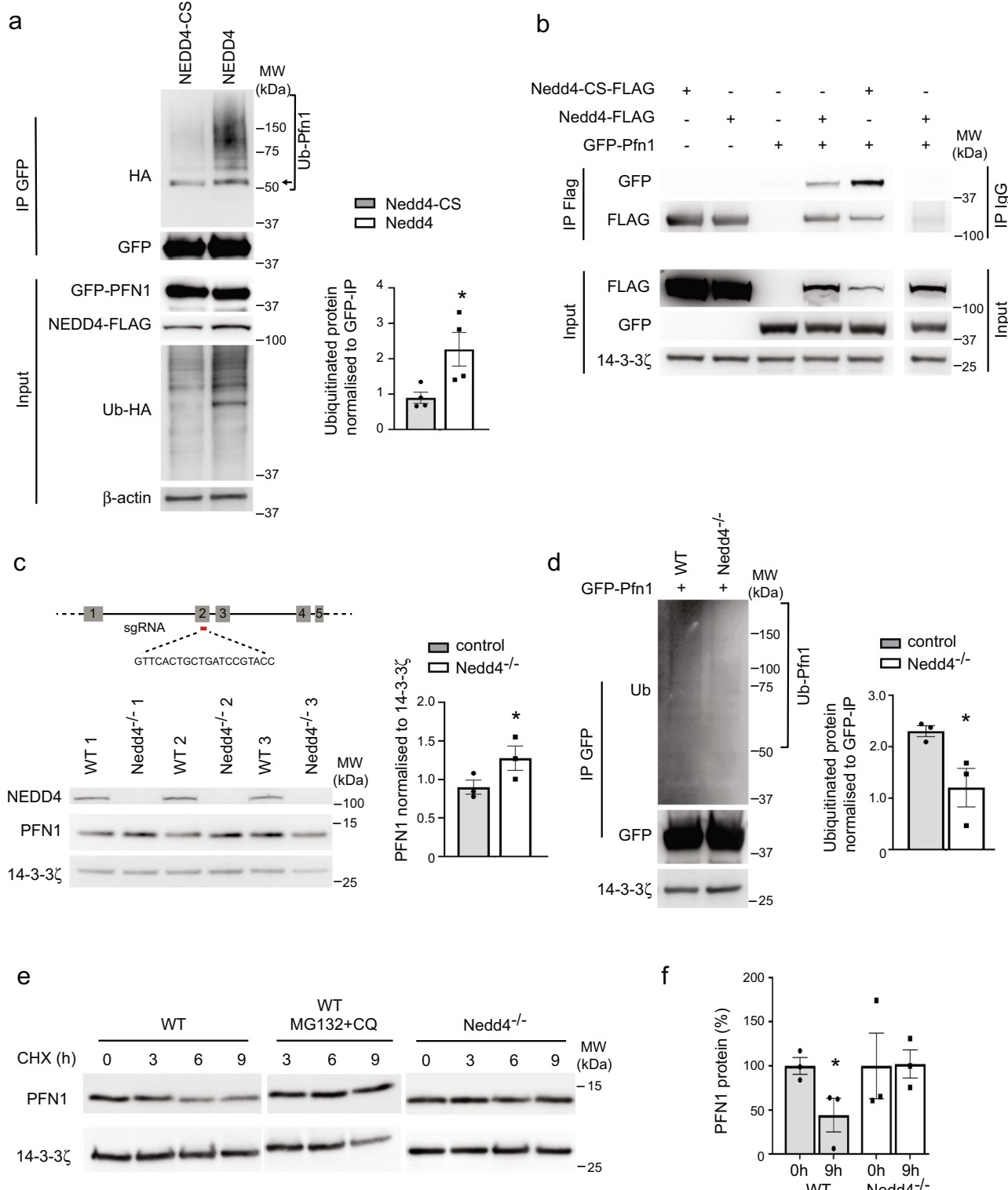

filamentous actin from monomeric G-actin we next quantitated the levels of G- and F-actin in NCCs. As expected, $Nedd4^{-/-}$ NCCs had an increased ratio of F:G actin compared to controls at steady-state levels (Fig. 7e). The ratio of F:G actin was also quantified in the Cyto D washout experiment. After CytoD treatment (0 min) the ratio of F:G actin was similar between control and $Nedd4^{-/-}$ NCCs (Fig. 7f). Consistent with our imaging analysis, after 180 min recovery there was a significant increase in the ratio of F:G actin in $Nedd4^{-/-}$ NCCs compared to

controls (Fig. 7f). Treatment of $Nedd4^{-/-}$ NCCs with $Pfn1$ siRNA also decreased the F:G actin ratio similar to controls (Fig. 7f).

To explore the role of ubiquitination on PFN1 function we next mapped ubiquitinated lysine residues. From a preliminary cell-based ubiquitination screen of lysine-to-alanine mutants we identified 3 lysine residues that altered the ubiquitination status of PFN1, including lysines 70, 108 and 127 (Fig. 8a). In addition to a reduction in larger ubiquitinated species above 75 kDa with the K108A mutant, quantitation identified a significant decrease to

**Fig. 4 NEDD4 ubiquitinates PFN1. a** In vivo ubiquitination assay with 293T cells transfected with HA-Ub, GFP-Pfn1 and Flag-Nedd4 or Flag-Nedd4-CS. Protein lysates were immunoprecipitated (IP) with GFP-trap beads. Ubiquitinated PFN1 (Ub-Pfn1) was probed with anti-HA antibody. Quantitation of the ubiquitinated species was performed from 4 independent experiments. Arrow indicates monoubuiqitinanted PFN1. Data are presented as mean $+/-$ SEM. *$P = 0.035$ as determined by type two Student's $t$ test. Uncropped images are provided as a Source data file. **b** 293T cells were transfected with various combinations of Flag-Nedd4, Flag-Nedd4-CS and GFP-Pfn1. NEDD4 and NEDD4-CS were precipitated from protein lysates with anti-Flag antibodies and compared to mouse IgG. Corresponding antibodies were used to recognize co-immunoprecipitated proteins. Uncropped images are provided as a Source data file. Blots are representative of 3 repeats. **c** Single-guide RNA (sgRNA) sequence designed to target exon 2 of *Nedd4* for gene knockout by CRISPR. Western blot of NCU10K NCC clones derived from single cells confirms complete KO of NEDD4 and altered abundance of PFN1. Quantitation of PFN1 levels from $n = 3$ biologically independent replicates. Data are presented as mean $+/-$ SEM. *$P = 0.038$ as determined by type-one Student's $t$ test. Uncropped images are provided as a Source data file. **d** In vivo ubiquitination assay with WT and *Nedd4*$^{-/-}$ NCCs transfected with GFP-Pfn1. Protein lysates were immunoprecipitated with GFP-trap beads and ubiquitination detected with anti-Ub antibodies. Quantitation of ubiquitin levels on immunoprecipitated GFP-PFN1 from 3 WT and 3 *Nedd4*$^{-/-}$ NCC lines. Data are presented as mean $+/-$ SEM. *$P = 0.048$ as determined by type-two Student's $t$ test. Source data are provided as a Source data file. **e** Cycloheximide (CHX) chase analysis of Pfn1 degradation in WT and *Nedd4*$^{-/-}$ NCCs. Blots are representative of 3 separate experiments. CQ, chloroquine. **f** Quantitation of cycloheximide chase assay from $n = 3$ biologically independent replicates. Data are presented as mean $+/-$ SEM. *$P = 0.043$ as determined by type one Student's $t$ test. Source data are provided as a Source data file.

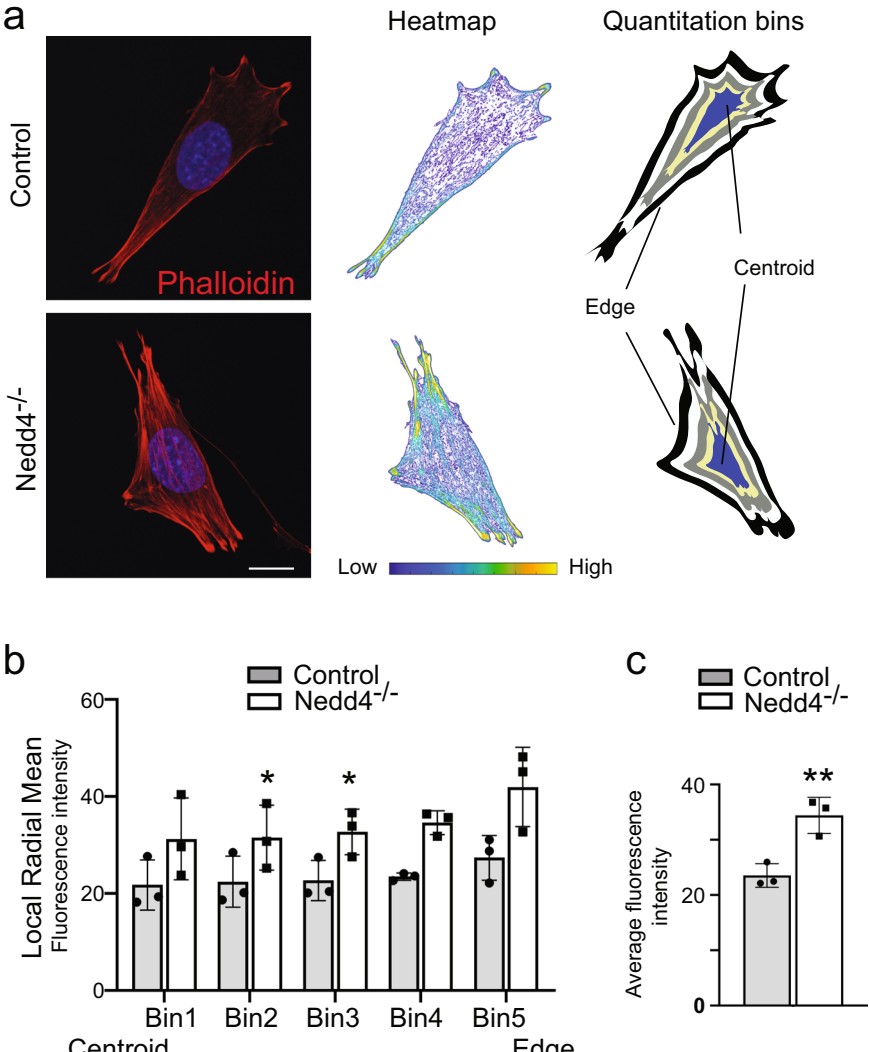

**Fig. 5 Quantitation of filamentous actin in *Nedd4*$^{-/-}$ NCCs. a** F-actin was stained in WT and *Nedd4*$^{-/-}$ NCCs with fluorescently conjugated phalloidin (red). Pseudocoloured heatmaps of actin intensity were used to measure actin levels by obtaining the Radial Mean Intensity in 5 separate bins from the centre to the edge. Images are representative of 10 individual cells per genotype. **b** Quantitation of actin levels in each of the bins from WT and *Nedd4*$^{-/-}$ NCCs. Data are presented as mean $+/-$ SEM. *$P = 0.049$ for Bin 3, $P = 0.0016$ for Bin 4, $P = 0.054$ for Bin 5 as determined by type two Student's $t$ test. **c** Quantitation of average F-actin levels in WT and *Nedd4*$^{-/-}$ NCCs. $n = 3$ biologically independent replicates. Scale bar $= 10\ \mu m$. Data are presented as mean $+/-$ SEM. **$P = 0.008$ as determined by type two Student's $t$ test. Source data are provided as a Source data file.

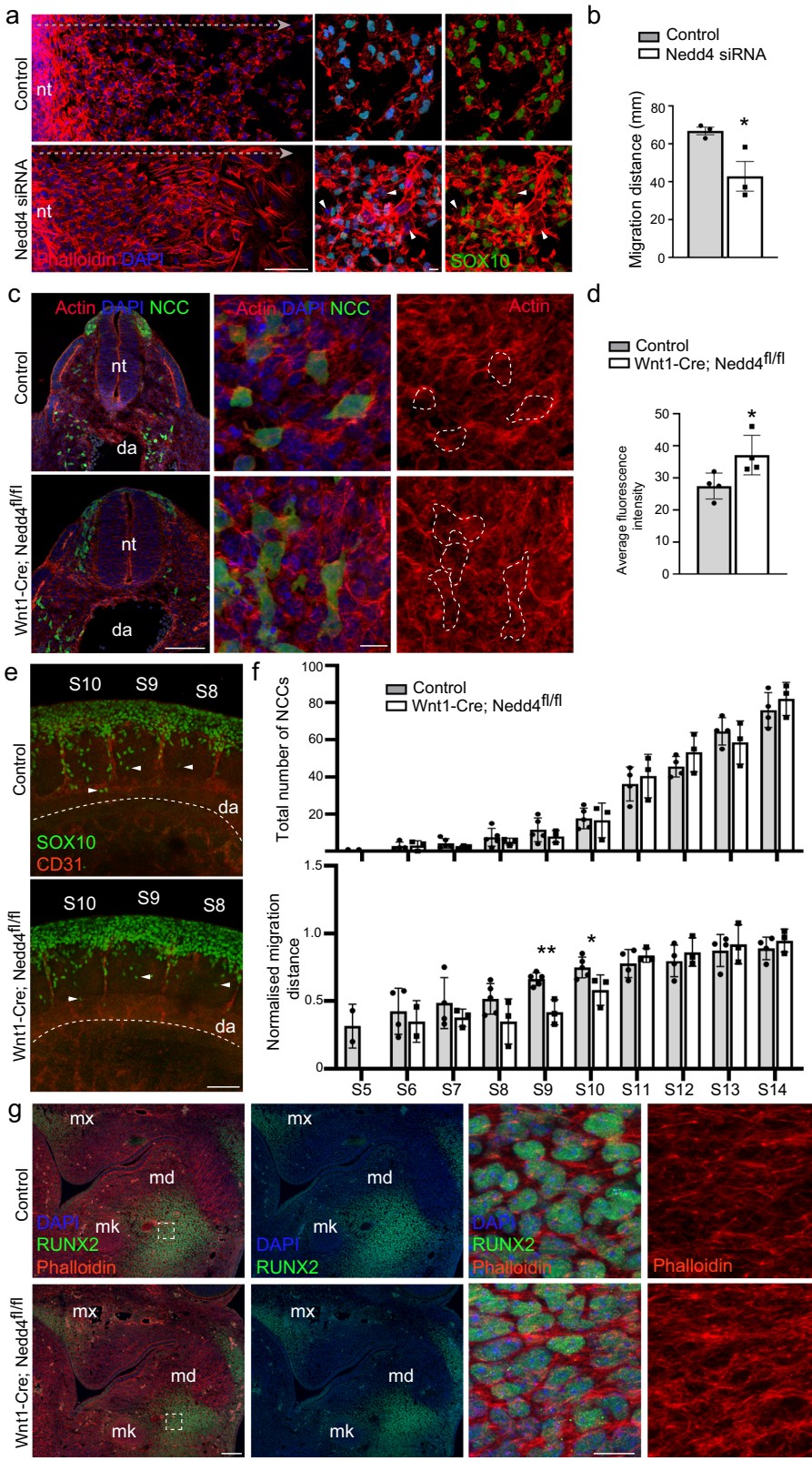

monoubiquitination when this lysine residue is replaced (Fig. 8b). In contrast, there was a significant increase to polyubuiqitination on K127A (Fig. 8b), which is consistent with replacement of this residue destabilising protein structure. Indeed, disease-causing mutations in PFN1 often lead to increased ubiquitination[22]. Compared to control GFP-PFN1, the K70A and K108A mutations also stabilized the PFN1 protein (Fig. 8c), similar to

that in *Nedd4*[−/−] NCCs (Fig. 4). Functionality of these lysine residues was also explored by analysing actin recovery with Cyto D washout experiments. F-actin stress fibre recovery and density were increased in NCCs expressing the K108A mutant protein, but not with the K70A mutant protein, therefore indicating that ubiquitination of lysine 108 plays a functional role in regulating actin polymerisation (Fig. 8d). We therefore conclude that Nedd4

**Fig. 6 Filamentous actin is increased in *Nedd4* knockdown and *Nedd4*$^{-/-}$ NCCs. a** Trunk neural tube (nt) explants from WT E9.5 embryos grown ex vivo treated with control or *Nedd4* siRNAs. NCCs were stained with anti-SOX10 antibodies (green) and phalloidin (red) and nuclei visualised with DAPI (blue). *Nedd4* knockdown led to reduced SOX10 and increased F-actin. Scale bar = 100 μm left; 10 μm right. Images are representative of 5 individual explants per treatment. **b** Quantitation of NCC migration over 12 h from trunk neural tube explants. *N* = 3 independent experiments of siRNA treated neural tube explants. Data are presented as mean +/− SEM. *P = 0.041 as determined by type two Student's *t* test. Source data are provided as a Source data file. **c** Representative low (left) and high magnification images (right) of transverse sections showing GFP + lineage traced migrating NCCs located next to the dorsal aorta (da) in control and *Wnt1-Cre; Nedd4*$^{fl/fl}$ E9.5 embryos. Sections were stained with anti-GFP antibodies (green) and phalloidin (red) and nuclei visualised with DAPI (blue). Dashed areas mark the regions used for quantitation. Scale bar = 100 μm left; 10 μm right. **d** Quantitation of actin content in GFP + NCCs located next to the dorsal aorta from control and *Wnt1-Cre; Nedd4*$^{fl/fl}$ E9.5 embryos. *N* = 4 biologically independent samples. Data are presented as mean +/− SEM. *P = 0.039 as determined by type two Student's *t* test. Source data are provided as a Source data file. **e** Representative images of migrating trunk NCCs within somites (S) 8–10 in control and *Wnt1-Cre; Nedd4*$^{fl/fl}$ E9.5 embryos. Whole embryos with 20–23 somites were stained with anti-SOX10 (green) and anti-CD31 antibodies (red) to recognise NCCs and blood vessels. Dashed line marks the ventral limit of the dorsal aorta (DA) which was used to normalise migration distance of pioneering NCCs (arrowhead). Scale bar = 100 μm. **f** Quantitation of NCC number and normalised migration distance in somites (s) 5–14 from *n* = 5 control embryos and *n* = 3 *Wnt1-Cre; Nedd4*$^{fl/fl}$ embryos. Data are presented as mean +/ − SEM. *P = 0.042 and **P = 0.002 as determined by type two Student's *t* test Source data are provided as a Source data file. **g** Frontal sections of *Wnt1-Cre; Nedd4*$^{fl/fl}$ and control embryos were immunostained for RUNX2 as an osteoblast marker (green), phalloidin (red) and nuclei visualised with DAPI (blue). Sections through the mandible and premaxilla are shown at low (left) and high magnification (right). Images are representative of 3 control and 3 *Wnt1-Cre; Nedd4*$^{fl/fl}$ embryos. mx, maxilla; md, mandible; mk, Meckel's cartilage. Scale bar = 100 μm left; 10 μm right.

promotes the ubiquitination of PFN1 on specific lysine residues to regulate its ability to control the dynamics of F-actin assembly and disassembly in NCCs.

## Discussion

Our current understanding of NCC development is biased toward the cell-autonomous roles of transcription factor networks acting downstream of morphogenetic signalling pathways[23]. However, recent work has also begun to explore the roles of post-translational modifications in this cell type[7,24]. Our own work has demonstrated that the ubiquitin ligase NEDD4 is required by NCCs to control craniofacial and peripheral nervous system development, providing a clear example that ubiquitination plays a critical role in this cell type. In this report we have applied a global proteomic strategy to explore how NEDD4 influences ubiquitination in NCCs and to identify NEDD4 substrates and molecular pathways through which NEDD4 controls NCC, craniofacial and peripheral nervous system development. We provide comprehensive data sets of genes, proteins and ubiquitinated lysine residues altered in the absence of NEDD4, and identify a suite of NEDD4 targets and interacting proteins. Our data demonstrate a role for NEDD4 in RNA transport, ribosome biogenesis and the actin cytoskeleton. We further uncover a role for NEDD4 as a regulator of the core actin-binding protein PFN1, positioning NEDD4 as a central protein in the actin polymerisation pathway.

Our global proteomics analysis uncovered several unexpected insights to ubiquitination and NEDD4 function in NCCs. Removal of *Nedd4* in NCCs led to a clear reduction of many ubiquitinated targets that led to a bimodal distribution in the heavy to light ratios of ubiquitination sites. While this was initially unexpected, global shifts in ubiquitination have also been documented in other proteomics studies in which E3 ligases have been removed[25,26] and in studies comparing proteosome inhibition to untreated cells[27]. We propose that a global shift in ubiquitin linkages arises due to NEDD4 having a predominant role in regulating NCC ubiquitination and NCC development. The lack of a corresponding shift in protein abundance is consistent with NEDD4 primarily forming mono and K63 ubiquitin linkages on target proteins which are associated with alterations of protein function rather than proteasomal degradation[28,29]. While *Nedd4* knockdown identified significant disruption to global K63 linkages, our data also uncovered significant disruption to K48 linkages, which are commonly associated with proteasomal degradation[4]. Putative K48 linkages regulating protein turnover

were uncovered for only 4 proteins that had corresponding increases in protein abundance and reduced ubiquitination in our global analyses. However, this is likely an under representation of K48 linked targets as altered protein abundance often induces regulatory feedback mechanisms on RNA expression to maintain homoeostasis, as has been documented for RNA binding proteins[30]. Indeed, in the case of PFN1 we found that a mild increase in protein abundance was only detectable by western blot and that this increase in protein level was coincident with reduced RNA expression. While a global reduction of K48 linkages in the absence of NEDD4 may be unexpected, recent studies looking at the yeast NEDD4 homologue, RSP5, reported a similar phenomenon where loss of this protein also resulted in strong effects on K48 and minimal effect on K63 linkages[31]. Based on the evidence that RSP5 and NEDD4 preferentially modify targets with single, or short chains of ubiquitin[32–35], as seen with our own analysis of PFN1, a general hypothesis is growing that ubiquitin chains on NEDD4 targets may be extended by other E3 ligases with alternative linkage preferences[36].

NEDD4 has been suggested to engage its substrates through interactions with target L/PPXY and "PY-like" motifs (generally S/VPXF)[31]. Analysis of our dataset revealed only 28% of proteins with affected ubiquitination sites contained L/PPXY or "PY-like" motifs (Supplementary Data 7). This finding is consistent with other high-throughput datasets examining NEDD4 targets, with only 30% of putative NEDD4 targets in rat containing L/PPXY motifs[18], and around 60% of putative RSP5 targets in yeast containing L/PPXY or "PY-like" motifs[31]. These findings therefore question the necessity of canonical interaction motifs for substrate ubiquitination by NEDD4. Previous studies also found that around 20% of reported substrates had direct interactions with NEDD4, whereas our MS approach identified 30% (81/276) of putative NEDD4 targets as direct interactors. Binding of NEDD4 to its target proteins has also been suggested to occur through adaptor proteins, such as the arrestin-related trafficking adaptors (ARTs)[37] and NEDD4 family-interacting proteins (NDFIP1 and 2)[16], which may help NEDD4 to recognize target proteins not identified as direct interactors in these studies. However, an alternative explanation is that a subset of proteins with altered ubiquitination following removal of E3 ligases are indirect. In our analyses we identified several other E3 ligases with affected ubiquitination site usage, altered total abundance, or as direct interactors of NEDD4. These observations suggest that changes in abundance or function of any E3 ubiquitin ligase may further affect the abundance or function of other components of

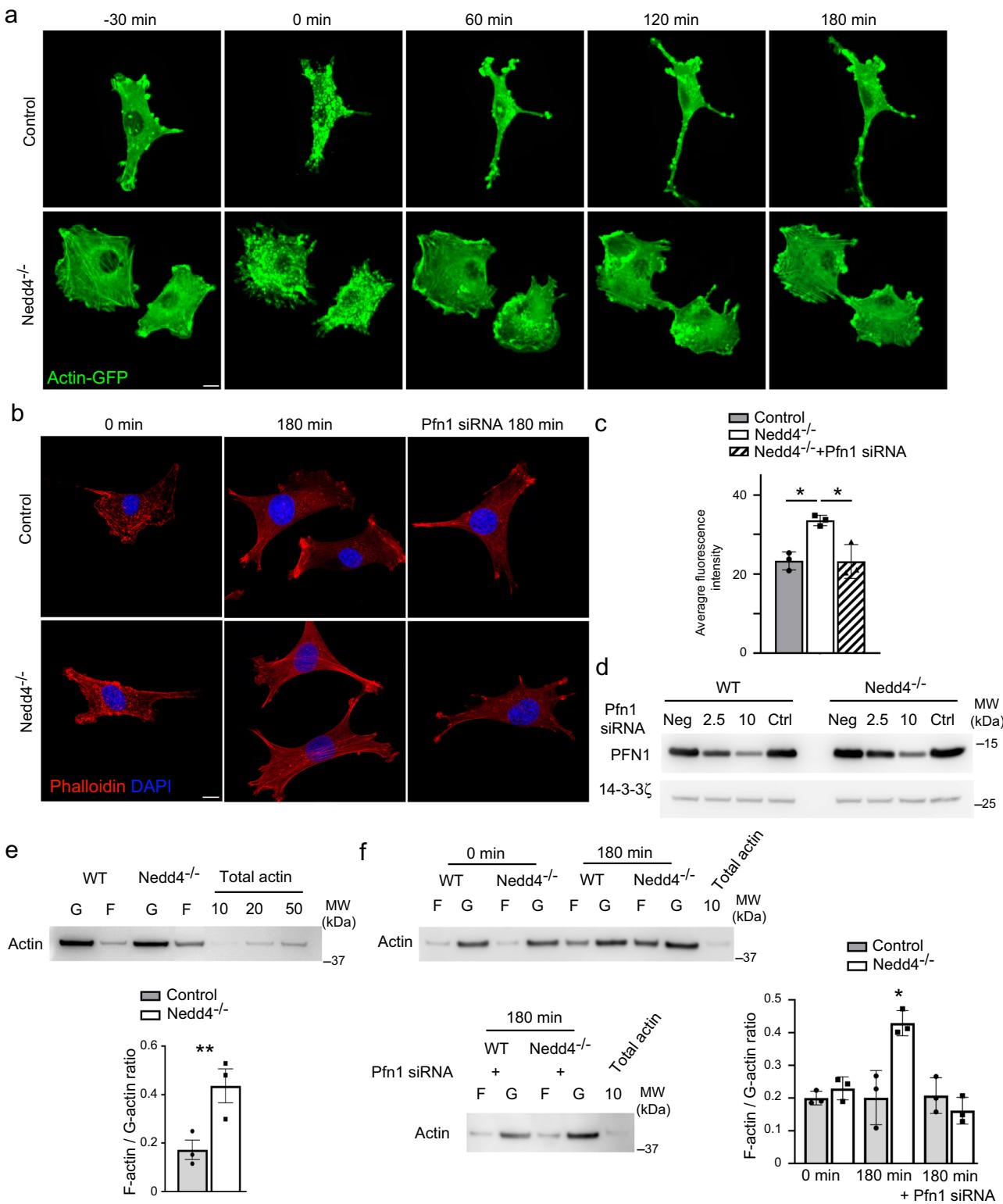

the ubiquitin-proteasome machinery. While this presents a challenge in determining which proteins are direct substrates of NEDD4, it also provides an exciting premise for NEDD4 playing a role at multiple levels of controlling ubiquitination; on one hand, as the primary enzyme involved in modifying substrates with ubiquitin moieties, while on the other, by orchestrating the ubiquitination of additional proteins through secondary E3 ligases. While multiple E3 ligase-containing complexes have been described for RING finger E3s[38], our finding that multiple HECT

domain-containing E3 ligases interact with NEDD4, suggests that additional E3 ligases with related functions can work together to imprint specific ubiquitin codes on target proteins with roles in specific cellular or developmental contexts. We propose that such complexes could also contribute to the large decrease in protein ubiquitination and bimodal distribution of ubiquitinated sites in NCCs after *Nedd4* knockdown.

A major aim of our global ubiquitinome analysis was to uncover NEDD4 targets that when aberrantly regulated may

**Fig. 7 NEDD4 regulates actin polymerisation. a** Representative time-lapse spinning disc confocal images of F-actin and stress fibre recovery after cytochalasin D washout in WT and Nedd4[−/−] NCCs expressing EGFP–Actin (green). Acquisitions started 30 min prior and immediately after washout. Images were acquired with the interval of 30 min after the washout. Scale bar = 10 μm. Images are representative of n = 3 biologically independent samples. **b** Representative images of F-actin (red) and stress fibre recovery after 180 min of cytochalasin D washout in WT and Nedd4[−/−] NCCs treated with control or Pfn1 siRNAs. Scale bar = 10 μm. Nuclei were visualised with DAPI (blue). **c** Quantitation of average F-actin level in WT and Nedd4[−/−] NCCs after 180 min cytochalasin D washout. N = 3 independent replicates. Data are presented as mean +/− SEM. *P = 0.0025 for WT v Nedd4[−/−] NCCs and P = 0.016 for Nedd4[−/−] NCCs v Nedd4[−/−] NCCs treated with Pfn1 siRNAs as determined by type two Student's t test. Source data are provided as a Source data file. **d** Abundance of Pfn1 protein detected with anti-Pfn1 antibodies in lysates of WT and Nedd4[−/−] NCCs treated with varying amounts of Pfn1 siRNAs (mM). Blots are representative of 3 repeats. Uncropped images are provided as a Source data file. **e** Representative western blot of F- and G-actin from WT and Nedd4[−/−] NCC protein lysates. Total load controls of known actin levels are shown. Quantitation of the F/G actin ratio is shown for n = 3 independent experiments. Data are presented as mean +/− SEM. *P = 0.03 as determined by type two Student's t test. Source data are provided as a Source data file. **f** Representative western blot of F- and G-actin from WT and Nedd4[−/−] NCC protein lysates at 0 min and 180 min after cytochalasin D washout with and without Pfn1 knockdown. Total load controls of known actin levels are shown. Quantitation of the F/G actin ratio is shown for n = 3 independent experiments. Data are presented as mean +/− SEM. *P = 0.012 as determined by type two Student's t test. Source data are provided as a Source data file.

---

underly the NCC defects identified in Nedd4[−/−] mice[7]. Providing strong support to the validity of our multi-omics approach is the observation that several proteins within our target list are previously identified substrates of NEDD4 (i.e. NDFIP and DVL2)[16,39]. Further credibility to the power of this approach comes from the finding that both of the targets tested with in vivo assays had enhanced ubiquitination in the presence of NEDD4. Taken together, our data therefore uncover a large number of previously unrecognized NEDD4 substrates and post-translational modifications that are essential for NCC development. This comprehensive list also provides candidate modifications that may have functional roles in other cell types.

Identification of PFN1 as a putative target in our global proteomics approach further uncovered a potential role for NEDD4 in regulating actin dynamics. The actin cytoskeleton is involved in numerous aspects of cellular function, with critical roles in migration, cellular division, cell signalling and differentiation. Central to the assembly and disassembly of filamentous actin are core actin-binding proteins that regulate the dissociation and polymerisation of actin monomers. PFN1 is a highly conserved actin-binding protein that facilitates nucleotide exchange from ADP to ATP to promote delivery of monomeric actin to the fast-growing end of new filaments, thereby enhancing polymerisation. Several of our findings demonstrate that NEDD4 negatively regulates the activity of PFN1 in NCCs: (1) Ubiquitination of PFN1 was significantly reduced in Nedd4 knockdown NCCs in our proteomics assay, (2) NEDD4, but not NEDD4-CS enhanced PFN1 ubiquitination in 293T cells, (3) Ubiquitination of GFP-PFN1 in Nedd4[−/−] NCCs was reduced, (4) PFN1 levels were increased in Nedd4[−/−] NCCs, (5) PFN1 turnover is reduced in Nedd4[−/−] NCCs, (6) Reduction of PFN1 in Nedd4[−/−] NCCs normalized actin levels and recovery after cytochalasin D treatment, and (7) Replacement of PFN1 lysine 108 with alanine decreased NEDD4-induced ubiquitination, stabilized the protein and increased the ability of PFN1 to regulate actin density. Our finding that PFN1 can be ubiquitinated in the absence of NEDD4 and that it is still ubiquitinated at lower levels after replacing single lysine residues also suggests that additional E3 ligases mediate ubiquitination of this protein and that multiple lysine residues are targeted for ubiquitination. In this regard, the E3 ligase CHIP has also been shown to ubiquitinate PFN1 to regulate its turnover in breast cancer cell lines[40] and this E3 ligase is also expressed in NCCs.

Consistent with NEDD4 ubiquitinating PFN1 to regulate its activity, we found that F-actin levels and stress fibre formation were altered in NCCs lacking NEDD4 and in cells expressing the ubiquitination resistant mutant GFP-PFN1-K108A. PFN1 has been proposed as a tumour suppressor in several cancers, with

reduced expression identified in breast, liver, bladder and pancreatic cancers[21,41–43]. Moreover, reduced levels of PFN1 have been found to promote migration and metastasis of breast cancer cells[21] while higher levels have been shown to decrease migration[44]. Increased levels of PFN1 have been suggested to inhibit migration by accelerating turnover of actin filaments[45]. In our own work, we find that altered PFN1 function and abundance as a result of Nedd4-deficiency accelerates actin polymerisation and density. This contrasting outcome likely comes from PFN1 levels only being marginally increased at steady-state levels in the absence of Nedd4, whereas accelerated F-actin turnover was induced with over 2-fold increases in protein level[45]. In addition to promoting protein turnover, our data suggest that ubiquitination of PFN1 also regulates its functional properties, as overexpression of GFP-PFN1 or expression of the GFP-PFN1-K70A stabilized mutant did not alter actin density when expressed in NCCs. PFN1 increases actin polymerisation by binding to G-actin and catalysing the exchange of ADP/ATP. Structural studies have found that actin primarily interacts with PFN1 on a surface comprising its β-sheets that are flanked by N- and C-terminal α-helices[46]. In contrast, a binding groove on the opposite side of the protein has been found to regulate interactions with ligands containing poly-proline motifs, including the actin regulators VASP and WAVE[47–49]. These alternate binding sites allow the formation of multi-protein complexes that are important for actin polymerisation. PFN1 also interacts with membrane phospholipids in a binding pocket that overlaps with the actin-binding site[50–52]. Thus, binding to actin and membrane phospholipids is in direct competition[51]. Lysine 108 is located on a loop region between the 6th and 7th β-sheets with close contacts to residues of the N- and C-terminal α-helices. As this lysine is thought to participate in binding to membrane phospholipids[53], post-translational modifications to this residue are likely to alter the balance between its interactions with phospholipids and actin. It will now be important to examine the role of ubiquitination in regulating PFN1-ligand interactions and its ability to catalyse the exchange of ADP/ATP for actin polymerisation. An interesting observation in previous work is that PFN1 is regulated by post translational modifications in a context-dependent manner. PFN1 is phosphorylated on several residues in endothelial cells following stimulation with vascular endothelial growth factor[54], and is differentially ubiquitinated in these cells under hypoxic stress[55]. Testing whether NEDD4 directs PFN1 ubiquitination under context-dependent conditions and/or in distinct compartments of the cell will be important to define its role in regulating PFN1 function and actin polymerisation.

Our previous analysis of Nedd4[−/−] embryos uncovered reduced expression of SOX10, aberrant NCC death and deficient

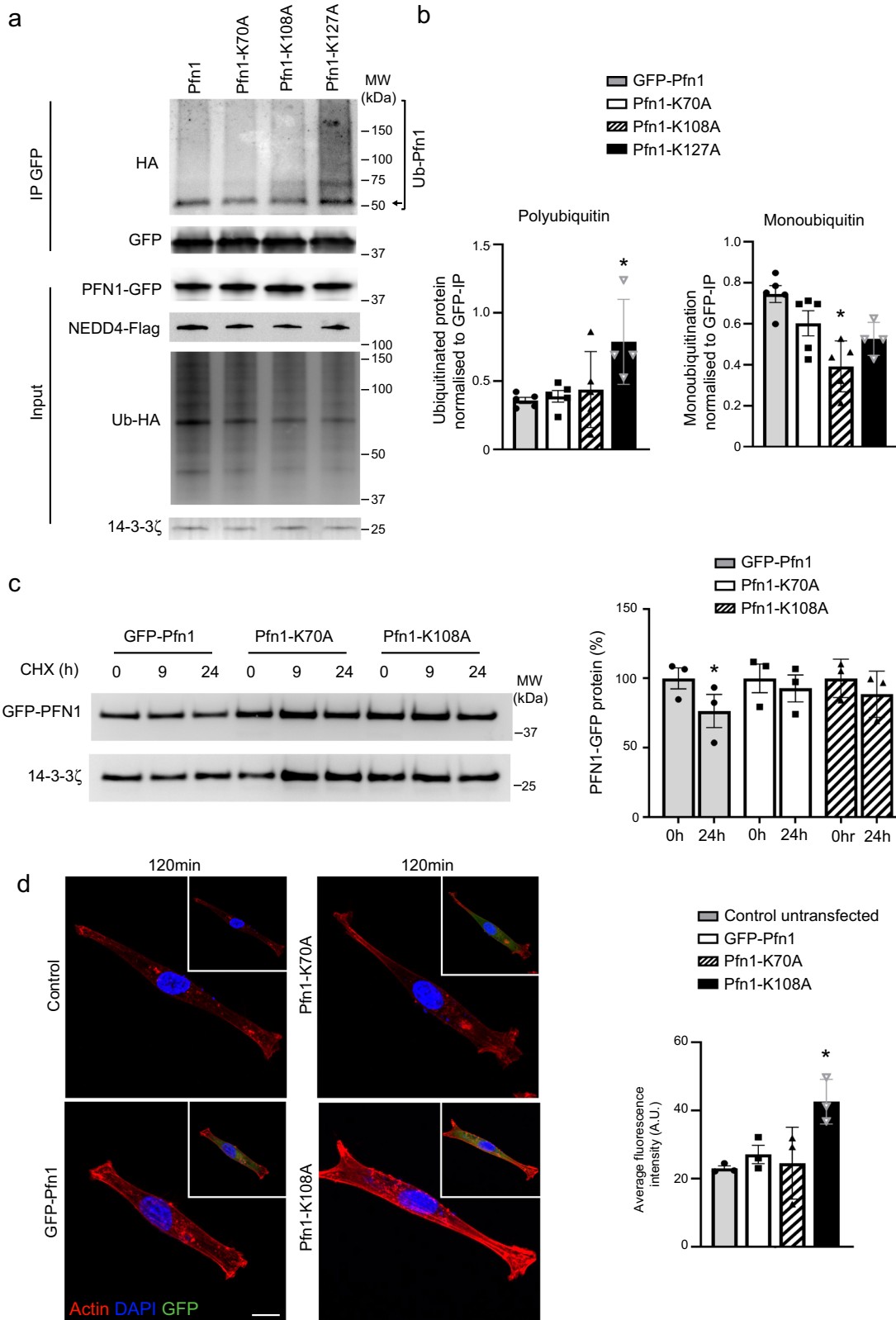

bone formation. Critical roles for the actin cytoskeleton have been identified in the regulation of core NCC transcription factors[56], NCC delamination[57], NCC migration[58] and in craniofacial morphogenesis[59]. Consistent with at least part of the *Nedd4*$^{-/-}$ NCC defect arising from alterations to actin polymerisation, morpholino knockdown of the actin-depolymerizing factor (ADF) in chick also had pronounced effects on SOX10

expression[56] and stabilization of actin filaments alters osteoblast differentiation from stromal stem cells[60]. It is also likely that additional Nedd4 substrates and disrupted biological pathways identified in this study play important roles in NCC development and the *Nedd4*$^{-/-}$ NCC phenotype. Outside of alterations to the cytoskeleton, our pathway analysis identified enrichment of several signalling pathways, including RAP1, PI3K-AKT, MAPK,

**Fig. 8 Ubiquitination of PFN1 regulates F-actin assembly. a** In vivo ubiquitination assay with 293T cells transfected with HA-Ub, Flag-Nedd4 and GFP-Pfn1, Pfn1-K70A, Pfn1-K108A or Pfn1-K127A. Protein lysates were immunoprecipitated with GFP-trap beads. Ubiquitinated PFN1 (Ub-Pfn1) was probed with anti-HA antibody. Uncropped images are provided as a Source data file. **b** Quantitation of ubiquitination levels from 5 independent experiments. Data are presented as mean +/− SEM. *$P = 0.017$ for polyubiquitination of Pfn1-K127A, $P = 0.0009$ for monoubiquitination of Pfn1-K108A and $P = 0.007$ for monoubiquitination of Pfn1-K127A as determined by type two Student's $t$ test. Source data are provided as a Source data file. **c** Cycloheximide (CHX) chase analysis of GFP-PFN1, PFN1-K70A and PFN1-K108A expressed in NCCs. Quantitation of protein levels after 24 h from 3 separate WT NCC lines expressing each PFN1 mutant, $n = 3$ independent experiments. Data are presented as mean +/− SEM. *$P = 0.049$ as determined by type one Student's $t$ test. **d** Representative images of F-actin (red) and stress fibre recovery after 120 min of cytochalasin D washout in control NCCs or NCCs expressing GFP-Pfn1, Pfn1-K70A and Pfn1-K108A (green). Nuclei were visualised with DAPI (blue). Scale bar = 10 μm. Quantitation of average F-actin level in NCCs after 120 min cytochalasin D washout. $n = 3$ independent experiments. Data are presented as mean +/− SEM. **$P = 0.007$ as determined by type one Student's $t$ test.

---

RAS and Notch. While NEDD4 has established roles in targeting proteins within the PI3K-AKT, RAS and Notch pathways[61], each of these pathways have been implicated in NCC development[62]. The phenotype of $Nedd4^{−/−}$ mice is indicative of severe, neural crest-specific developmental defects, which in humans are collectively referred to as neurocristopathies. Recent advances in understanding the aetiology of neurocristopathies have also revealed dysregulated ribosome biogenesis as a direct causative agent in two well-studied examples, Treacher Collins Syndrome and Diamond Blackfan anaemia[63]. Both of these disorders arise from mutations that disturb ribosome biogenesis, leading to stabilization of P53 and consequently, G1 cell cycle arrest and apoptosis of NCCs[64–66]. Intriguingly, while the dependence of NCCs on proper regulation of ribosome dynamics and function remains poorly understood, it is known that the expression profiles of individual ribosomal proteins during vertebrate development are heterogeneous in terms of both cell-type and tissue-specificity[67]. Furthermore, disruption to just a single ribosomal protein produces tissue-restricted axial skeletal phenotypes, through dysregulated translation of a subset of mRNAs without impact on global translation. Our process enrichment analysis suggests that NEDD4 specifically ubiquitinates multiple protein components of complexes intimately involved in ribosome function and RNA transport. A number of these proteins including MAGOH, EIF4A3 and Importin beta-1 have been shown independently to influence the proliferation and survival of neural stem cells and cells derived from neural crest[68–70]. The ability of NEDD4 to target multiple substrates simultaneously further suggests that NEDD4 may act as a central hub in coordinating multiple signalling pathways and cellular processes to regulate NCC development.

Finally, a notable correlate to our finding that NEDD4 inhibits PFN1 function in NCCs are the contrasting roles of these proteins in cancer and neurodegeneration. While reduced levels and loss of function mutations in PFN1 are linked with cancer progression[21] and neurodegeneration[22], increased levels of NEDD4 are suggested to have oncogenic properties[71] and have also been documented in neurodegeneration[72]. Our discovery of an important mechanism regulating PFN1 stability and function therefore provides insight toward the treatment of its pathological roles in several disease settings.

In summary, we report a systematic analysis of the NEDD4 ubiquitinome in NCCs. Our findings uncover a previously unrecognized role for NEDD4 in integrating regulatory processes that govern NCC development and identifies a crucial role for NEDD4 in regulating PFN1 and the actin cytoskeleton.

## Methods

**Mouse lines**. All experiments were carried out in accordance with ethical guidelines of the SA Pathology and University of South Australia Animal Ethics Committees. To obtain embryos of defined gestational ages, mice were mated in the evening, and the morning of vaginal plug formation was counted as embryonic day (E) 0.5. Pregnant dams were humanely euthanized at relevant days post vaginal plug detection by $CO_2$ inhalation and cervical dislocation. To remove $Nedd4$ specifically in NCCs we crossed $Wnt1\text{-}Cre;Nedd4^{fl/+}$ males to $Nedd4^{fl/fl}$ females as previously described[6,35]. To lineage trace neural crest cells, $Wnt1\text{-}Cre;Nedd4^{fl/+}$ males were crossed to $Z/EG$ females. $Nedd4^{fl/+}$ mice were originally supplied by Hiroshi Kawabe[35], $Wnt1\text{-}Cre$ mice were originally supplied by Tim Thomas[73] and $Z/EG$ mice were originally supplied by Andas Nagy[74]. All mice were maintained on a mixed Sv129 - C57BL/6 background on a 12 h dark/light cycle at 20–22 °C with 40–60% humidity and fed a normal chow diet. The sex of embryos used for our studies was not determined.

**Cell and tissue culture**. The immortalized neural crest cell line NCU10K was obtained from Perry Bartlett[7] and used in quantitative proteomics and microarray experiments. For microarray experiments, NCU10K cells were grown in DMEM (21051024 Thermo Fisher) supplemented with 10% foetal calf serum (10099141 Thermo Fisher). For quantitative proteomics NCU10K cells were grown in DMEM minus arginine and lysine (A33822 Thermo Fisher). For heavy isotope labelled media, DMEM was supplemented with L-Lysine-$^{13}C_6$ $^{15}N_2$-HCl (K8, 100 μg/mL final, 89988 Thermo Scientific) and L-Arginine-$^{13}C_6$ $^{15}N_4$ (R10, 12.5 μg/mL final, 88433 Thermo Scientific). For light isotope labelled media L-Lysine-2HCl (K0, 88429 Thermo Fisher) and L-Arginine-HCl (R0, 89989 Thermo Scientific) were used at the above concentrations. All media contained 10% dialysed foetal calf serum (236400044 Thermo Fisher). Incorporation of heavy amino acids surpassed 98% after 6 cell doublings and arginine to proline conversion was <2%. For cell-based ubiquitination assays, 293T cells originally obtained from ATCC (CRL-3216) were grown in DMEM supplemented with 10% foetal calf serum. Plasmid transfections were carried out with Lipofectamine 3000 (L3000015 Thermo Fisher). CRISPR/Cas9 targeting of the $Nedd4$ locus was performed with pSpCas9(BB)-2A-GFP (px458) (Addgene plasmid # 48137). Forward (caccGTTCACTGCT-GATCCGTACC) and reverse primers (aaacGGTACGGATCAGCAGTGAAC) were annealed and cloned into the BbsI site. Efficient Cas9 cleavage at the desired site was determined with GeneArt™ Genomic Cleavage Detection Kit (Life Technology) using primers flanking the cut site (CTGGATAAGACAGAAGATGG and GATCTGGATTCAATTCCTAGC). GFP positive single cells expressing either the cloning vector (WT) or the Nedd4 targeting vector ($Nedd4^{−/−}$) were cloned with NEDD4 disruption validated by western blot. Tanswell migration assays were performed by plating 10,000 cells into the top of 8 um transwells (3422 Corning) containing DMEM with DMEM supplemented with 10% foetal calf serum placed in the bottom well. After 24 h the transwells were rinsed with 1X phosphate-buffered saline (PBS, 14190144 Thermo Fisher), cells fixed with 4% paraformaldehyde (PFA) (158127 Merck) and stained with crystal violet. Cells remaining in the top well were removed with cotton swabs. Images were taken on Olympus SZX10 microscope with Openlab software (Improvision). Trunk neural tubes were isolated from E9.5 embryos for the culture of primary NCCs[75]. siRNA transfections were performed using Lipofectamine RNAimax (13778150 Thermo Fisher) with $Nedd4$ Stealth siRNA (GCCAGAGAGUGGUUCUUCCUCAUCU) (MSS206986 Thermo Fisher) or low GC control siRNA (12935200 Thermo Fisher) for 48 h before harvesting. Images of siRNA treated cells were taken on an Olympus CKX41 microscope. For primary neural tubes siRNA was transfected 4 h after plating[7]. Images of actin-GFP transfected cells and primary neural tube explants were acquired using a CellVoyager™ CV1000 spinning disc microscope (Yokogawa) with time-lapsed imaging of actin recovery captured every 10 min. CV1000 software (Yokogawa) was used to generate images and movies. Translation shut-off experiments were performed with 50 μg/ml of cycloheximide (C4859 Merck) in normal growth media. NCCs were seeded in 12-well plates with equal number of cells in each well. Following treatment with cycloheximide for the indicated time points cells were harvested for immunoblotting analysis on LAS-4000 with Multi Gauge software (Fuji Films). Actin depolymerization was performed with 2 mM cytochalasin D (C8273 Merck) in normal growth media. Treatment of transfected NCCs started 24 h after transfection. F-actin and stress fibre recovery was stimulated with 3 washing steps with normal growth medium.

**Protein extraction and digestion for MS**. NCCs in 10 × 10 cm plate were washed twice with ice-cold PBS and lysed in 10 mL lysis buffer containing 20 mM HEPES

(15630080 Thermo Fisher) at pH 8.0, 8 M urea (U5378 Merck), 1 mM sodium orthovanadate (S6508 Merck), 2.5 mM sodium pyrophosphate (71505 Merck), 1 mM β-glycerophosphate (G9422 Merck). Samples were sonicated using a Sonics Vibra-cell micro-tip sonicator (3 mm probe). Lysates were cleared by centrifugation at $20,000 \times g$ for 15 min and transferred to a fresh 50 mL tube (Falcon). Protein concentration was estimated using the EZQ fluorescence-based protein assay (R33200 Thermo Fisher) and 5 mg of each lysate (i.e. 5 mg K8R10 *Nedd4* siRNA and 5 mg K0R0 control siRNA) combined (final volume ~8 mL). Proteins were reduced with dithiothreitol (4.5 mM final) for 1 h at room temp and carbamidomethylated with iodoacetamide (10 mM final) 15 min at room temp. The sample was then diluted to 2 M urea with 20 mM HEPES pH 8.0 and digested overnight at room temperature with trypsin-TPCK (1:31 w/w) (TRTPCK Worthington). Following digestion, samples were acidified with formic acid (F0507 Merck) and desalted using a 360 mg C18 Sep-Pak Classic cartridge (WAT054955 Waters). C18 cartridges were conditioned with 5 ml of 100% MeCN (34851 Merck), followed by 5 ml of 50% MeCN, 0.1% FA (302031 Merck), and finally 20 ml of 0.1% trifluoroacetic acid (TFA). Sample was loaded onto the conditioned C18 cartridge, washed with 15 ml of 0.1% TFA, and eluted with 6 ml of 50% MeCN, 0.1% FA. Desalted samples were frozen to −80 °C in a 50 mL tube and dried to completeness overnight in an RVC 2-18 CDplus rotational vacuum concentrator (Martin Christ) connected to an Alpha 2-4 LDplus freeze drier set to 0.01 mbar vacuum at −70 °C (Martin Christ).

**Basic reversed-phase (RP) chromatography.** Off-line basic RP fractionation[76] was performed with a custom-manufactured Zorbax 300 Extend-C18 column (9.4 × 250 mm, 300 Å, 5 μm, Agilent). This column was coupled to a Waters 1525 Binary HPLC pump complete with 2489 UV/vis detector and fraction collector. Dried peptides were resuspended in 1.8 ml of basic RP solvent A (2% MeCN, 5 mM ammonium formate, pH 10 (70221 Merck)), and loaded into a 2 mL loop. We used a 64 min basic RP liquid chromatography (LC) method consisting of an initial increase to 8% solvent B (90% MeCN, 5 mM ammonium formate, pH 10) at 1.1% B/min followed by a 38-min linear gradient (0.5% B/min) from 8% solvent B to 27% B and successive ramps to 31% B (1% B/min), 39% B (0.5% B/min) and 60% B (3% B/min). The entire LC separation was carried out at 3 ml/min. Upon sample injection, 80 basic RP fractions were collected in 48 s intervals (2.4 ml/fraction) into 10 mL polypropylene tubes (Falcon). Following separation, the fractions were pooled following a non-contiguous pooling strategy into eight total fractions in 50 mL tubes (essentially 10 sub-fractions combined to make one final fraction following the rule $8*[1–10] + n$ where $n =$ the final fraction number [1–8]).

**Anti-K-ε-GG antibody cross-linking and ubiquitin remnant enrichment.** Antibody cross-linking and ubiquitin remnant enrichment was performed following previous methods[76] with an antibody bead slurry provided in the PTMScan® ubiquitin remnant motif (K-ε-GG) kit (5562 Cell Signaling Technology). Washed beads were crosslinked by resuspending the antibody beads in 1 ml of 20 mM dimethyl pimelimidate (DMP) (D8388 Merck) at room temperature. The reaction was quenched with 200 mM ethanolamine, pH 8.0 (E9508 Merck) followed by incubation in 200 mM ethanolamine for 2 h at 4 °C with rotation. After quenching, the antibody beads were washed three times and finally resuspended in immuno-affinity purification buffer (IAP) (50 mM MOPS, pH 7.2 (M1254 Merck), 10 mM sodium phosphate (342483 Merck), 50 mM NaCl (S9888 Merck)) buffer. Each of the eight peptide fractions was resuspended in 1.5 mL IAP. 1450 μL of each peptide fraction were transferred to low bind microfuge tubes (Eppendorf) with 50 μL reserved for "pre-enrichment" protein identification and abundance measurements by LC-MS/MS. 55 μL of the crosslinked K-ε-GG antibody beads was added to each 1450 μL peptide fraction and incubated overnight at 4 °C with rotation. K-ε-GG remnant beads were washed four times with 1.5 ml of ice-cold PBS followed by elution of the ubiquitin remnant peptides with 0.15% trifluoroacetic acetic. Purified peptides were desalted using homemade C18 StageTips[77], eluted with 50 μL of 50% acetonitrile, 0.1% FA, dried to completeness and resuspended in 2% acetonitrile, 0.1% FA.

**LC-MS/MS analysis.** Fractionated peptides were analysed by LC-MS/MS using a Q Exactive mass spectrometer (Thermo Scientific, Bremen, Germany) coupled online with a RSLC nano uHPLC (Ultimate 3000, Thermo Scientific, Bremen, Germany). Samples were loaded on a 100 μm, 2 cm pepmap 100 trap column (164946 Thermo Fisher) in 2% MeCN, 0.1% formic acid at a flow rate of 15 μl/min. Peptides were eluted and separated at a flow rate of 300 μl/min on a RSLC nanocolumn 75 μm × 50 cm, pepmap100 C18, 3 μm 100 Å pore size (164946 Thermo Fisher), using 60 min linear gradient 2–24% acetonitrile for pre-enrichment samples and 30 min gradient for enriched peptides. The eluent was nebulised and ionised using the Thermo nano electrospray source with a distal coated fused silica emitter (New Objective, Woburn, MA, USA) with a capillary voltage of 1.7–2.1 kV. The Q Exactive instrument was operated in the data-dependent mode to automatically switch between full-scan MS and MS/MS acquisition. Survey full-scan MS spectra ($m/z$ 375–1600) were acquired in the Orbitrap with 70,000 resolution ($m/z$ 200) after accumulation of ions to a $3 \times 10^6$ target value with maximum injection time of 120 ms. Dynamic exclusion was set to 15 s. The 12 most intense multiply charged ions ($z \geq 2$) were sequentially isolated

and fragmented in the octopole collision cell by higher-energy collisional dissociation (HCD) with a fixed injection time of 60 ms 17,500 resolution and AGC target of $1 \times 10^5$ counts. 2.5 Da isolation width was chosen. Underfill ratio was at 10% and dynamic exclusion was set to 15 s. Typical mass spectrometric conditions were as follows: spray voltage, 2 kV; no sheath and auxiliary gas flow; heated capillary temperature, 275 °C; normalized HCD collision energy 27% and subjected to LC-MS analysis.

**MS data analysis.** MS data were analysed using MaxQuant[78] version 1.3.0.5 and searched against the mouse UniprotKB database (version 2013_09 containing 75,952 entries). Also included was a list of 248 common laboratory contaminants provided by MaxQuant. For the search parameters, enzyme specificity was set to trypsin with the maximum number of missed cleavages set to 2. Precursor mass tolerance was set to 20 ppm for the first search (used for nonlinear mass re-calibration) and 6 ppm for the main search. Variable modifications included were oxidized methionine, Gly-Gly modified lysines, and N-terminal protein acetylation. Carbamidomethylation modification of cysteines was fixed. The false discovery rate (FDR) for peptide, protein, and site identification was set to 1%; the minimum peptide length was set to 7 and the filter-labelled amino acid, peptide requantification and match between runs functions were enabled. All eight fractions from both biological replicates were searched together using the experiment template to define them as separate experiments with eight fractions each. Pre-enrichment fractions (total) were searched separately to ubiquitin remnant enriched fractions. The match between runs function was enabled in both cases. Ubiquitination site identification and quantification information were obtained from the MaxQuant Gly-Gly sites and evidence data. Protein group identification and quantification were obtained from the proteinGroups data. Contaminants and reversed database hits were removed prior to data processing. Significant changes in pre-enrichment protein abundance were determined using Perseus (statistical software available with MaxQuant) and based on significance B with a Benjamini–Hochberg corrected FDR < 5%. For evaluating diglycine peptide assignments, any peptide with a ubiquitin remnant on a C-terminal lysine that had a better than 50% localization probability was excluded. Identification and quantification of protein groups from the "total" samples was obtained from the proteinGroups data. The normalised H/L values for both "total" and ubiquitin remnant enriched datasets are reported (log-transformed to base 2). In all cases, the leading razor protein is the reported protein ID. Where proteins in a protein group cannot be distinguished in a data both variants are listed.

**Microarray preparation and data analysis.** Microarrays were performed at the Adelaide Microarray Centre (SA Pathology/University of Adelaide, South Australia, Australia). All microarray analyses were performed on 3 biological replicates. RNA was prepared from a single well of a 24-well plate with Qiagen RNeasy microcolumns (Qiagen). RNA sample integrity was measured using an Agilent Bioanalyzer (Agilent Technologies, Santa Clara, CA, USA) prior to microarray analysis. 300 ng samples of high-quality RNA were prepared for microarray using the GeneChip WT Expression Kit and Affymetrix Terminal Labelling and Control Kit (900670 Thermo Fisher) as per the manufacturers' protocols. Affymetrix GeneChip Gene 1.0 ST Mouse Arrays were used for gene expression analysis. After overnight hybridisation, the microarrays were washed and stained using Affymetrix Hybridisation Wash and Stain reagents on an Affymetrix GeneChip Fluidics Station 450. Scanning was performed on a GeneChip 3000 7G Scanner with Command Console software. Affymetrix raw array data were processed using Partek Genomics Suite software (Partek). Briefly, .cel files were imported using RMA background correction, Partek's own GC content correction, and mean probe summarization. Differential gene expression was assessed by ANOVA with the $p$-value adjusted using step-up multiple test correction to control the false discovery rate. When comparing gene expression and proteomic data, the leading razor protein UniprotKB ID was used to obtain Refseq nucleotide identifiers using the UniProt retrieve/ID mapping function.

**NEDD4 co-immunoprecipitation and LC-MS/MS analysis.** HEAVY and LIGHT SILAC NCU10K cells were lysed in non-denaturing lysis buffer (20 mM sodium phosphate buffer pH 7.5, 150 mM NaCl, 5% Glycerol (G5516 Merck), 1% NP40 (NP40 Merck), Complete Protease Inhibitor cocktail (CO-RO Merck), 2.5 mM sodium pyrophosphate, 1 mM β-glycerophosphate, 1 mM sodium orthovanadate) and precleared with protein G sepharose (P3296 Merck). 50 μL of protein G dynabeads (10004D Thermo Fisher) preincubated with 4 μg of anti-NEDD4 (BD Bioscience) or isotype-matched IgG control mouse antibody (Jackson) in PBS with 0.1% Triton-X 100 (PBST) (X100 Merck) were added to 1 mg of lysate for 2 h at 4 °C. Dynabeads were washed twice with PBST and immunoprecipitated material eluted in 2x SDS load buffer and separated on a 4–12% NuPAGE polyacrylamide gel (WG1203 Thermo Fisher). The gel was stained with coomassie G250 (B0149 Merck) and 13 equal-sized bands cut for analysis. Each band was submitted to in-gel tryptic digest and peptides analysed using a Q Exactive mass spectrometer (Thermo Fisher) coupled online with a RSLC nano uHPLC (Ultimate 3000, Thermo Fisher) and data interpreted using MaxQuant.

**Immunoprecipitation and western blot**. Cells were lysed in NP-40 buffer (137 mM NaCl, 10 mM Tris pH 7.4 (108319 Merck), 10% glycerol, 1% NP-40, 2 mM sodium fluoride and 2 mM sodium vanadate) containing Complete Protease Inhibitors. For immunoprecipitation, the supernatants were precleared with mouse Ig-coupled Sepharose beads for 30 min at 4 °C. The precleared lysates were incubated for 2 h at 4 °C with 2 µg/mL of anti-Flag (F1804 Merck) or control mouse IgG absorbed to protein G-Sepharose. The sepharose beads were washed 3 times with lysis buffer before being boiled for 5 min in SDS-PAGE sample buffer. For purification of GFP-fusion proteins GFP-Trap beads were used following the manufacturers' protocols (gta-10 Chromotek)[79]. The GFP-Pfn10 vector containing mouse Pfn1 fused to GFP was a gift from Michael Davidson (Addgene plasmid # 56438). Single lysine residues were replaced by alanine by inserting modified gene fragments (IDT) into the XhoI and XbaI sites of the GFP-Pfn10 vector. The pEYFP-C1-Kif16b vector was a kind gift from Marino Zerial. Fractionation of F and G actin was performed following the manufacturers protocols (BK037 Cytoskeleton). Protein samples were separated by SDS-PAGE and transferred to PVDF membrane (88585 Thermo Fisher) with known amounts of actin used for quantitation in F:G actin fractionation experiments. Membranes were blocked in 5% skim milk powder in tris buffered saline (TBS, 50 mM Tris pH 7.5, 150 mM NaCl) with 0.1% Tween 20 (P1379 Merck). The following primary antibodies were used: anti-actin (AAN02 Pierce, 1 in 1000), anti-SOX10 (ab155279 Abcam, 1 in 4000), anti-ERBB3 (Ab-1289 AAT Bioquest, 1 in 150), anti-FNDC3B (ab135714 Abcam, 1 in 500), anti-ZEB2 (NBP1-82991 Novus Biologicals in 1000), anti-NEDD4 (611481 BD, 1 in 3000), anti-ubiquitin P4G7 (838703 Biolegend, 1:1000), anti-ubiquitin K63 (5621 Cell Signalling Technologies, 1:1000), anti-ubiquitin K48 (8081 Cell Signalling Technologies, 1:1000), anti PFN1 (3237 Cell Signalling Technologies, 1:1000), anti 14-3-3zeta C-16 (sc1019 Santa Cruz, 1:1000)[80] and mouse IgG isotype control (Jackson, used at 4 µg/mg of lysate).

**Immunohistochemistry**. Cells and tissue sections were fixed in 4% PFA (158127 Merck). Embryos were cryosectioned at 12 µm. Cryosections and fixed cells were blocked in 10% DAKO serum-free blocking solution (X0909 DAKO) in PBST. The following primary antibodies and stains were used: anti-SOX10 (AF2864 R&D, 1:200), anti-RUNX2 (12556 Cell Signalling, 1:250), Alexa-fluor 594 phalloidin (A12381 Life technologies, 1:1000). The following secondary antibodies were used at 1:200 dilution: anti-rabbit AlexaFluor 488 (A21206 Thermo Fisher); anti-rabbit AlexaFluor 555 (A31572 Thermo Fisher); anti-mouse AlexaFluor 647 (A31571 Thermo Fisher); anti-mouse AlexaFluor 488 (A21202 Thermo Fisher); anti-rat AlexaFluor 488 (A21208 Thermo Fisher). Tissue sections and cells were mounted in Prolong Diamond Anti-fade with DAPI (P36962 Thermo Fisher). Images were acquired using a Zeiss LSM 800 confocal microscope and ZEN (blue) software.

**Quantitation and statistics**. Actin quantitation on fluorescent confocal images was performed in Matlab with custom made code obtained from Rafael Aldabe[20]. Radial Mean Intensity was used to quantify protein distribution independent of cell shape and size. Cell morphology was converted into a circular distribution by applying to each local intensity pixel a geometric transformation, which depends only on the distance of the pixel to the centroid and the closest edge of the cell. These circles were split into five separate bins in which fluorescence intensity was measured and averaged from each cell analysed. In vivo migration was quantitated by measuring the distance (m) in each furthest NCC in each somite and normalising this to the distance (d) to the ventral limit of the dorsal aorta (Supplementary Fig. 8). Densitometry of all western blot data and single-cell tracking was performed with ImageJ. All data are presented as mean ± SEM. All experimental studies were analysed using Student's *t* test with a *P* value of <0.05 was considered to be statistically significant. Pathway enrichment was performed using DAVID ontology enrichment tool[81] in which *P* values were determined using Fisher's exact test with adjustments using Bonferroni, Benjamini and FDR methods. FDR adjusted *p*-value based on Benjamini–Hochberg Step-Up FDR-controlling Procedure were used for microarray analyses.

**Reporting summary**. Further information on research design is available in the Nature Research Reporting Summary linked to this article.

## Data availability

The data that support this study are available from the corresponding author upon reasonable request. Microarray data have been deposited in the NCBI Gene Expression Omnibus (GEO)[82] with the dataset identifier GSE197631 (Gene expression changes in NCU10K cells lacking Nedd4). The mass spectrometry proteomics data have been deposited to the ProteomeXchange via PRIDE[83] partner repository with the dataset identifier PXD024103 (SILAC MS in NCU10K cells lacking Nedd4). Source data are provided with this paper.

## Code availability

Matlab codes and quantitative distribution software used in this study were written and published by Rafael Aldabe[20] who provided them upon request. This software is available on request from Rafael Aldabe (raldabe@unav.es).

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

## Acknowledgements
The authors thank Xiangjun Xu for help with genotyping. This work has been supported by the following grants: National Health and Medical Research Council (1144008 and 1144004; Q.S.) (1002863, 1020755; S.K.), Max Planck Society (H.K.), German Research Foundation (KA3423/3-1; H.K.), JSPS KAKENHI (JP20K07334; H.K.), Ohsumi Frontier Science Foundation (H.K.), Takeda Science Foundation (H.K.) and The Uehara Memorial Foundation (H.K.).

## Author contributions
I.L., P.M., S.W. R.D., S.K., N.L.H. and Q.S. conceived the project. I.L., P.M., S.K., H.K. and Q.S. wrote and edited the manuscript. J.W. and P.M. carried out bioinformatic analyses. I.L., P.M., S.W., N.A., G.S., C.M. and O.K. performed all experiments.

## Competing interests
The authors declare no competing interests.

**Additional information**

