## [Peer Review File · Nature Communications]

Reviewers' comments:

Reviewer #1 (Remarks to the Author):

The authors provide a comprehensive dataset of ubiquitome, global proteome, and transcriptome for Nedd4 knockdown cells together with protein network analysis in the manuscript. The information in this manuscript, especially the result of ubiquitome, is extremely informative for the Nedd4 biology. Given that Nedd4 is ubiquitously expressed and is involved in numbers of biological phenomenon, this paper will appeal to a broad spectrum of readers. This manuscript also demonstrates that such a global approach is required to understand the roles of an E3 ubiquitin ligase in a certain cell type. Also the manuscript is focused on screenings of substrates of an E3 ubiquitin ligase, which perfectly fits to the scope of Nature Communications, covering "fields that aren't represented by a dedicated Nature research journal". I fully support this manuscript to be published in Nature Communications when authors can improve several minor issues.

1) Figure 4F, reduction of K63 in Nedd4 KD cells is obvious. It is, from my point of view, a striking result. This indicates that a single E3 ligase, Nedd4, contributes to most of the K63 chain formation in the cell. However, I do not see a similar level of the change in Western blotting with the anti-K48 antibody. They should provide more convincing result for this. Authors should also provide the Western blotting result for an internal control (e.g. actin or GAPDH) or the image of nitrocellulose membrane stained with Ponceau or MemCode to be sure that the loading amounts are comparable between Nedd4 KD and control cells.

2) The way they provide the list of genes/proteins in the supplemental tables is difficult to follow. They should highlight the genes/proteins that shows changes in protein expression level (Supplemental table 1) in top hits in the ubiquitin remnant study (Supplemental table 2).

3) The left panel of Figure 6 is confusing. I assume that the top band is the signal from FLAG tagged Fndc3b and the lower intense bands are from Nedd4. However, judging from the image, two antibodies (i.e. anti-FLAG and anti-Nedd4) seem to be incubated with the single nitrocellulose membrane. If this is the case, how can authors be sure that the top band is not the cross reactivity from the anti-Nedd4 antibody and the bottom one is not from the anti-FLAG antibody? The blot should be done separately. Also there is a faint band with the same molecular weight as Nedd4 in the second lane but not in the forth lane. Does it mean that the wild type Nedd4 binds to FLAG-Fndc3b while the inactive mutant does not? It would cause a big problem to interpret the result if it was the case.

In the right panel of "image enhanced", I do not see specific bands in the left lane. Authors should repeat this experiment to provide a more convincing result.

Authors should also provide the information how many times they repeated this and other biochemical experiments. At least, one biochemical experiment should be repeated twice and authors should show representative data.

I appreciate that numbers of potential substrates in Supplemental Figure 4 have been already reported as substrates of Nedd4 in other cell lines. However, in order to reinforce the manuscript, ubiquitination assays should be done for at least three more newly identified substrates in Figure 6.

4) In Figure 5, authors should include two more information. In Figure 5A, they should include the protein names that showed decreases in K- ϵ -GG peptides and increases in protein abundance. Obvious question is if they are transmembrane proteins or not. In Figure 5D, authors should provide a list of names of proteins with more than one K- ϵ -GG peptides reduced in the Nedd4 KD cells. This criteria will help to identify the substrate of E3 ligases as shown before (Yamazaki et al., EMBO J, 32, 524). Authors should cite this paper too.

5) Legends for the Supplemental Tables are not complete. Authors need to provide more information. For example, what each column indicates? For instance, what are slices 1 to 8, what is PEP in Supplemental Table1? How do they normalize the values (e.g. Log₂ H/L)? Given that Nature Communications is the journal that appeal to a broad spectrum of readers, authors need to provide information so that non-experts can understand each individual value.

6) Supplemental table 3 should provide the protein name in the list. What is the description in column R?

7) In Supplemental table 4, did authors quantify the fold changes of ubiquitin remnant peptides in column W based on the signal height or the signal area? In any cases, authors need to provide the original values from each group (i.e. KD and control) for each experiment (not the averaged value but the original raw data).

8) Page 7, line 13, a period is missing after "Fig".

9) In most of the cases, K48 and K63 are major polyubiquitin chains; sometimes these two chain types cover more than 50% of the total polyubiquitin chains. However, authors conclude that overall polyubiquitin level is not changed (page 7, line 13). Please explain why.

10) Page 15, line 11, authors should provide the exact company name rather than "BD".

11) Page 18, line 16, an abbreviation IAP buffer appears all of sudden and it is explained line 18 in the same page. This should be first explained in line 16.

12) Page 22, line 4, they provide two company names for protein G dynabeads. They need to make it clear whether they use the beads from Thermo Scientific or Pierce.

--

Reviewer #2 (Remarks to the Author):

In this manuscript McCarthy et al. used quantitative mass spectrometry (MS) to identify proteins whose ubiquitination is affected after Nedd4 knockdown. They used NCU10K cell line and knocked down Nedd4 using RNAi. They analyzed changes in protein, and ubiquitination abundance by quantitative MS.

While the authors raise broad and interesting questions in their abstract and introduction, their data barely provide any novel, clear insight into these questions. This appears like a typical proteomic exercise (of modest depth) and the manuscript is almost entirely based on descriptive presentation of the data. They have identified a list of proteins that are up- or down-regulated in Nedd4 knockdown cells. There is no way to know if they are direct substrates of Nedd4. The figures contain very little useful and informative data. For example, entire figure 1 is a classical proteomics workflow that contains absolutely no new information. Figure 3 is a hairball without conveying anything interesting and useful; there are so many connecting lines that it is even impossible to see the connections. The same is true for figure 5. There is no novel findings confirmed in the manuscript, except validation of a single protein presented in figure 6. The experiment presented in Figure 6 fails to clearly substantiate their conclusions- there is no clear decrease in Fndc3b ubiquitination. Furthermore, it is known that overexpression of ubiquitin ligases can promiscuously ubiquitinate proteins that are not necessarily their physiological substrates (Danielsen et al., Mol Cell Proteomics.

2011 Mar;10(3):M110.003590). Thus, their approach to simultaneously overexpress target protein, Nedd4, and ubiquitin is prone to the above mentioned problem. A better approach would be test if knockdown of Nedd4 leads to reduced ubiquitination of endogenous Nedd4 targets.

Also, in Figure 4E, K33, K48 and K63-linked ubiquitination is decreased to the exact same extent. However, their validation experiment (presented in Figure 4F) show very different results for K48 and K63 linkages. While K63 linkages appear to show decrease in Nedd4 knockdown cells, there is no appreciable difference for K48 linkages. It is unclear how many replicates were performed for these experiments and how reproducible these blots are?

Minor points:

- The statement "...peptides subjected to ubiquitin remnant motif (K-e-GG) enrichment..." needs original references describing this approach (Kim et al., Mol Cell. 2011 Oct 21;44(2):325-40; Wagner et al., Mol Cell Proteomics. 2011 Oct;10(10):M111.013284).
- Almost none of Nedd4 targets are confirmed here. Thus, at best, they should be identified as "putative" Nedd4 targets so that it is clear that they are not validated targets.

--

Reviewer #3 (Remarks to the Author):

This manuscript used SILAC quantitative proteomic approaches to identify proteins and ubiquitination sites differentially regulated by an E3 ligase Nedd4 in neural crest cells (NCCs). The authors also conducted the gene expression profiling experiments and found that there was a significant difference between the mRNA and protein expression level upon Nedd4 knockdown. The authors performed bioinformatic analyses and found the pathways that the differentially regulated proteins are involved in. The findings from these experiments are consistent with previous experiments in morphological changes of NCCs upon knockdown of Nedd4. There are no experiments to demonstrate that the proteins or the ubiquitination events regulated by Nedd4 are the ones that are critical for NCC and craniofacial development. Therefore, this manuscript does not provide sufficient novelty and evidence to support the main conclusion. Please see below for details.

1. Further experiments are required to demonstrate that Nedd4 regulates craniofacial development or morphological change of NCC through the differentially regulated proteins or ubiquitinated proteins. Simple tabulation and bioinformatic analyses of proteomic data at protein level and at the ubiquitination level do not support the major conclusions of the manuscript.

2. The authors state that their analyses revealed a novel role for Nedd4 in cell-type specific regulation of protein translation machinery. But no further experiments support this conclusion.

3. In the authors' previous experiments, they found an aberrant increase in NCC death, perturbed NCC stem-cell like properties and a profound reduction in craniofacial bone and peripheral neurons in mice lacking Nedd4. In the current work, the authors concluded that how neural crest cell development and formation of the craniofacial skeleton and peripheral nervous system may be influenced by Nedd4. However, this work did not provide the direct signaling pathways that Nedd4 regulates these processes.

4. It is surprising that two major ubiquitination chain linkages (K48 and K63) are significantly reduced from the proteomic data while the Western blot showed that there is no significant change in K48 linked ubiquitination after knocking down Nedd4. The discrepancy should be explained in detail. Blotting with total ubiquitin is needed to evaluate the alteration in the overall ubiquitination level. The standard deviation for Figure 4E should be provided. In addition, it is hard to read the labels in Figure 4F.

5. In Figure 6, the authors stated that "Western blots probing the IP material for myc-tagged ubiquitin shows a band at a slightly higher molecular weight than Fndc3b (133kDa) that is virtually undetectable in cells cotransfected with the catalytically dead Nedd4 (Fig. 6)." However, the Western blot did not show any distinguishable difference between the samples transfected with wild type or catalytically dead Nedd4 mutant. In addition, proper labels should be provided in Figure 6. It is hard to tell which antibody is really used for Western blotting of the left panel. No loading control was provided for these panels. The band below Nedd4 should also be labeled. A better experiment for this is to knock down Nedd4 instead of overexpressing the wild type and mutant Nedd4.

6. The authors discovered a bimodal distribution of ubiquitinated peptides in the absence of Nedd4. Reasons causing this bimodal distribution should be provided. For general readers, it is hard to believe that reduction of a single E3 ligase can cause such a significant effect on protein ubiquitination.

7. In Figure 5A, how is the number of the protein groups obtained? In Figure 5B, based on the scatter plot, the 3-fold increased and 3-fold reduced points do not follow the criteria described in the main text.

8. The authors mentioned that their experiments provided a foundation for studies examining direct versus indirect Nedd4 targets, disentangling mono-ubiquitinated targets from poly-ubiquitinated,

and determining exactly which poly-ubiquitin linkages are directed by Nedd4 on any given target. Detailed experiments should be provided to demonstrate this principle.

9. In the motif search analysis, the authors searched the known motifs, PPXY and PY-like. Are there any new motifs in the differentially regulated proteins? How do these motifs affect the ubiquitination events regulated by Nedd4?

10. The labels for the ratio in Figure 2D and 2E need to be corrected. The criteria for the change in the abundance at the protein level should be explicitly stated to show the differentially regulated proteins in Figure 2E.

11. In the second panel of Supplementary Figure 2, both the non-modified and the ubiquitinated Ebb3 are increased after knocking down Nedd4. The authors should provide an explanation for this. In addition, the ratio for WB should also be provided.

12. Figure 3 could be incorporated in Figure 2.

13. There are some typos. For example:

On page 5: over all

In the legend of Figure 5: unaffected ubiquitin remnant peptides are shown in "purple". Purple should be "black".

In Table 1, the authors pointed out that "Reactome results are colour coded to match Fig. 7." But the colors are not matched.

Response to Reviewer comments - NCOMMS-15-11765A-Z:

Reviewer #1

The authors provide a comprehensive dataset of ubiquitome, global proteome, and transcriptome for Nedd4 knockdown cells together with protein network analysis in the manuscript. The information in this manuscript, especially the result of ubiquitome, is extremely informative for the Nedd4 biology. Given that Nedd4 is ubiquitously expressed and is involved in numbers of biological phenomenon, this paper will appeal to a broad spectrum of readers. This manuscript also demonstrates that such a global approach is required to understand the roles of an E3 ubiquitin ligase in a certain cell type. Also the manuscript is focused on screenings of substrates of an E3 ubiquitin ligase, which perfectly fits to the scope of Nature Communications, covering "fields that aren't represented by a dedicated Nature research journal". I fully support this manuscript to be published in Nature Communications when authors can improve several minor issues.

1) Figure 4F, reduction of K63 in Nedd4 KD cells is obvious. It is, from my point of view, a striking result. This indicates that a single E3 ligase, Nedd4, contributes to most of the K63 chain formation in the cell. However, I do not see a similar level of the change in Western blotting with the anti-K48 antibody. They should provide more convincing result for this. Authors should also provide the Western blotting result for an internal control (e.g. actin or GAPDH) or the image of nitrocellulose membrane stained with Ponceau or MemCode to be sure that the loading amounts are comparable between Nedd4 KD and control cells.

Response: Western blot for K63, K48 and total ubiquitin with matching load controls are now included in the revised manuscript.

2) The way they provide the list of genes/proteins in the supplemental tables is difficult to follow. They should highlight the genes/proteins that shows changes in protein expression level (Supplemental table 1) in top hits in the ubiquitin remnant study (Supplemental table 2).

Response: All supplemental tables have now been revised to clarify which proteins meet our criteria for being included as potential Nedd4 targets. We also include a table in the main figures with putative targets annotated within the actin cytoskeleton pathway.

3) The left panel of Figure 6 is confusing. I assume that the top band is the signal from FLAG tagged Fndc3b and the lower intense bands are from Nedd4. However, judging from the image, two antibodies (i.e. anti-FLAG and anti-Nedd4) seem to be incubated with the single nitrocellulose membrane. If this is the case, how can authors be sure that the top band is not the cross reactivity from the anti-Nedd4 antibody and the bottom one is not from the anti-FLAG antibody? The blot should be done separately. Also there is a faint band with the same molecular weight as Nedd4 in the second lane but not in the forth lane. Does it mean that the wild type Nedd4 binds to FLAG-Fndc3b while the inactive mutant does not? It would cause a big problem to interpret the result if it was the case.

In the right panel of "image enhanced", I do not see specific bands in the left lane. Authors should repeat this experiment to provide a more convincing result.

Authors should also provide the information how many times they repeated this and other biochemical experiments. At least, one biochemical experiment should be repeated twice and authors should show representative data.

I appreciate that numbers of potential substrates in Supplemental Figure 4 have been already reported as substrates of Nedd4 in other cell lines. However, in order to reinforce the manuscript, ubiquitination assays should be done for at least three more newly identified substrates in Figure 6.

Response: Most if this is no longer applicable. Since our previous submission we have performed ubiquitination assays on over 5 of the putative Nedd4 targets to validate our proteomics approach. Two of the putative targets with known roles in regulating the cytoskeleton are included in the current manuscript with follow up functional studies in cells and tissues from Nedd4 mutant mouse embryos.

4) In Figure 5, authors should include two more information. In Figure 5A, they should include the protein names that showed decreases in K-ε-GG peptides and increases in protein abundance. Obvious question is if they are transmembrane proteins or not. In Figure 5D, authors should provide a list of names of proteins with more than one K-ε-GG peptides reduced in the Nedd4 KD cells. This criteria will help to identify the substrate of E3 ligases as shown before (Yamazaki et al., EMBO J, 32, 524). Authors should cite this paper too.

Response: This information is supplied in the supplemental tables.

5) Legends for the Supplemental Tables are not complete. Authors need to provide more information. For example, what each column indicates? For instance, what are slices 1 to 8, what is PEP in Supplemental Table1? How do they normalize the values (e.g. Log2 H/L)? Given that Nature Communications is the journal that appeal to a broad spectrum of readers, authors need to provide information so that non-experts can understand each individual value.

Response: Additional information is provided in the material and methods.

6) Supplemental table 3 should provide the protein name in the list. What is the description in column R?

Response: This is no longer applicable.

7) In Supplemental table 4, did authors quantify the fold changes of ubiquitin remnant peptides in column W based on the signal height or the signal area? In any cases, authors need to provide the original values from each group (i.e. KD and control) for each experiment (not the averaged value but the original raw data).

Response: Area under the curve was used to quantitate peptide abundance. Raw data will be uploaded to PRIDE upon acceptance for publication.

8) Page 7, line 13, a period is missing after "Fig".

Response: This has been corrected.

9) In most of the cases, K48 and K63 are major polyubiquitin chains; sometimes these two chain types cover more than 50% of the total polyubiquitin chains. However, authors

conclude that overall polyubiquitin level is not changed (page 7, line 13). Please explain why.

Response: This comment has been removed and no longer applicable in the new manuscript.

10) Page 15, line 11, authors should provide the exact company name rather than "BD".

Response: BD refers to Becton Dickinson Biosciences and will be amended for final submission.

11) Page 18, line 16, an abbreviation IAP buffer appears all of sudden and it is explained line 18 in the same page. This should be first explained in line 16.

Response: IAP buffer is commercially available from NEB. This will be amended for final submission.

12) Page 22, line 4, they provide two company names for protein G dynabeads. They need to make it clear whether they use the beads from Thermo Scientific or Pierce.

Response: This has been amended.

Reviewer #2

In this manuscript McCarthy et al. used quantitative mass spectrometry (MS) to identify proteins whose ubiquitination is affected after Nedd4 knockdown. They used NCU10K cell line and knocked down Nedd4 using RNAi. They analyzed changes in protein, and ubiquitination abundance by quantitative MS.

While the authors raise broad and interesting questions in their abstract and introduction, their data barely provide any novel, clear insight into these questions. This appears like a typical proteomic exercise (of modest depth) and the manuscript is almost entirely based on descriptive presentation of the data.

Response: We agree that the original submission was heavily biased to the description of data obtained from our global proteomics and transcriptomics approaches. We now provide validation of several novel Nedd4 targets and functional analysis of the role for Nedd4 ubiquitination in actin polymerisation using Nedd4 knockout cell lines and Nedd4 mutant mouse embryos.

They have identified a list of proteins that are up- or down-regulated in Nedd4 knockdown cells.

Response: Our manuscript combines quantitative analysis of the total proteome with quantitative transcriptomic data and quantitative proteomic analysis of the ubiquitinome. These datasets are directly comparable and complement each other. Indeed, with few exceptions, most previous proteomics experiments looking at the ubiquitinome fail to measure total protein levels. As such, there is no way to know that a reported change in a specific ubiquitination event is not due to an abundance change in cognate protein. The approach we have taken to identify ubiquitination events affected by loss of Nedd4 does not suffer this caveat and provides one of the most robust datasets currently available using these technologies. The quality of the data is exemplified by the fact that nearly 50% of the candidate substrates are Nedd4 interacting proteins. Given we are working in a cell type for which no previous proteomic analysis of Nedd4 has been published, this level of correlation

with previously reported Nedd4 interacting proteins further supports the validity of our dataset. In our updated manuscript we provide further validation of the proteomics data and functional analysis of the mechanisms by which Nedd4 controls neural crest cell development.

There is no way to know if they are direct substrates of Nedd4.

Response: This is an unfortunate and valid point of most global approaches to identify E3 substrates. In our dataset we find that around 50% of putative targets are known or novel interactors of Nedd4, providing confidence that they may be direct substrates. In the revised manuscript we have validated two of the putative targets identified from our proteomics approaches, that were also found to interact with Nedd4. We also raise this point in the discussion. In work not included in this manuscript we have validated additional targets identified from our proteomics approaches, including Dvl2, Dvl3 and Mib1. Notably, however, Dvl1 is not a Nedd4 substrate in neural crest, albeit that it has been suggested as a substrate in other cell types. These latter examples have not been included in the current manuscript due to our focus on the actin cytoskeleton. We are happy to share this data with the reviewers if they would like to see this.

The figures contain very little useful and informative data. For example, entire figure 1 is a classical proteomics workflow that contains absolutely no new information. Figure 3 is a hairball without conveying anything interesting and useful; there are so many connecting lines that it is even impossible to see the connections. The same is true for figure 5.

Response: Due to our change in focus, figures 3 and 5 have now been removed and are no longer applicable. Figure 1 is now included as a supplemental figure to provide the reader with a visual description of the general protocol used for our proteomics analysis.

There is no novel findings confirmed in the manuscript, except validation of a single protein presented in figure 6. The experiment presented in Figure 6 fails to clearly substantiate their conclusions- there is no clear decrease in Fndc3b ubiquitination.

Response: We now provide 4 new data figures confirming Profilin 1 as a novel substrate of Nedd4 in neural crest cells.

Furthermore, it is known that overexpression of ubiquitin ligases can promiscuously ubiquitinate proteins that are not necessarily their physiological substrates (Danielsen et al., Mol Cell Proteomics. 2011 Mar;10(3):M110.003590). Thus, their approach to simultaneously overexpress target protein, Nedd4, and ubiquitin is prone to the above mentioned problem. A better approach would be test if knockdown of Nedd4 leads to reduced ubiquitination of endogenous Nedd4 targets.

Response: While the Danielsen paper presents a detailed analysis of global ubiquitination in U2OS cells stably expressing HA-tagged ubiquitin, it does not include overexpression of a ubiquitin ligase. We are not aware of studies that identify off-target ubiquitination by an E3 ligase overexpressed in a heterologous cell system. Regardless of this, we show clear differences in target ubiquitination between cells expressing wild type Nedd4 or Nedd4-CS. Our attempts to immunoprecipitate endogenous Profilin 1 from WT and KO Nedd4 neural crest cells was unsuccessful, as were our attempts to perform TUBE assays for this protein. To circumvent these issues we purified GFP-tagged Profilin 1 from neural crest cells to show mild reduction in ubiquitination in KO cells. Furthermore, we find that Profilin 1 turnover is

affected in the absence of Nedd4, confirming it as a novel substrate in this cell type.

Also, in Figure 4E, K33, K48 and K63-linked ubiquitination is decreased to the exact same extent. However, their validation experiment (presented in Figure 4F) show very different results for K48 and K63 linkages. While K63 linkages appear to show decrease in Nedd4 knockdown cells, there is no appreciable difference for K48 linkages. It is unclear how many replicates were performed for these experiments and how reproducible these blots are?

Response: Western blot for K63, K48 and total ubiquitin with matching load controls are now included in the revised manuscript. These have all been performed more than 3 times.

Reviewer #3

This manuscript used SILAC quantitative proteomic approaches to identify proteins and ubiquitination sites differentially regulated by an E3 ligase Nedd4 in neural crest cells (NCCs). The authors also conducted the gene expression profiling experiments and found that there was a significant difference between the mRNA and protein expression level upon Nedd4 knockdown. The authors performed bioinformatic analyses and found the pathways that the differentially regulated proteins are involved in. The findings from these experiments are consistent with previous experiments in morphological changes of NCCs upon knockdown of Nedd4. There are no experiments to demonstrate that the proteins or the ubiquitination events regulated by Nedd4 are the ones that are critical for NCC and craniofacial development. Therefore, this manuscript does not provide sufficient novelty and evidence to support the main conclusion. Please see below for details.

1. Further experiments are required to demonstrate that Nedd4 regulates craniofacial development or morphological change of NCC through the differentially regulated proteins or ubiquitinated proteins. Simple tabulation and bioinformatic analyses of proteomic data at protein level and at the ubiquitination level do not support the major conclusions of the manuscript.

Response: We now provide substantive evidence confirming Profilin 1 as a novel substrate of Nedd4 in neural crest cells. We show that this affects the formation and turnover of filamentous actin in neural crest cell lines and in vivo.

2. The authors state that their analyses revealed a novel role for Nedd4 in cell-type specific regulation of protein translation machinery. But no further experiments support this conclusion.

Response: This is no longer applicable. The focus of the modified manuscript is on the role of Nedd4 regulating Profilin 1 and the actin cytoskeleton.

3. In the authors' previous experiments, they found an aberrant increase in NCC death, perturbed NCC stem-cell like properties and a profound reduction in craniofacial bone and peripheral neurons in mice lacking Nedd4. In the current work, the authors concluded that how neural crest cell development and formation of the craniofacial skeleton and peripheral nervous system may be influenced by Nedd4. However, this work did not provide the direct signaling pathways that Nedd4 regulates these processes.

Response: The current manuscript provides new data confirming a central role for Nedd4 in regulating actin dynamics, through regulating the turnover of a core actin binding protein, Profilin 1. We show perturbed actin formation in Nedd4 KO neural crest cell lines and in

Nedd4 KO neural crest cells in vivo. Notably, reducing the levels of Profilin 1 rescues actin polymerisation defects in Nedd4 KO cells.

4. It is surprising that two major ubiquitination chain linkages (K48 and K63) are significantly reduced from the proteomic data while the Western blot showed that there is no significant change in K48 linked ubiquitination after knocking down Nedd4. The discrepancy should be explained in detail. Blotting with total ubiquitin is needed to evaluate the alteration in the overall ubiquitination level. The standard deviation for Figure 4E should be provided. In addition, it is hard to read the labels in Figure 4F.

Response: Western blot for K63, K48 and total ubiquitin with matching load controls are now included in the revised manuscript. These have all been performed more than 3 times.

5. In Figure 6, the authors stated that "Western blots probing the IP material for myc-tagged ubiquitin shows a band at a slightly higher molecular weight than Fndc3b (133kDa) that is virtually undetectable in cells cotransfected with the catalytically dead Nedd4 (Fig. 6)." However, the Western blot did not show any distinguishable difference between the samples transfected with wild type or catalytically dead Nedd4 mutant. In addition, proper labels should be provided in Figure 6. It is hard to tell which antibody is really used for Western blotting of the left panel. No loading control was provided for these panels. The band below Nedd4 should also be labeled. A better experiment for this is to knock down Nedd4 instead of overexpressing the wild type and mutant Nedd4.

Response: Analysis of Fndc3b is no longer applicable. We now provide 4 new data figures confirming Profilin 1 as a novel substrate of Nedd4 in neural crest cells. This includes the generation of Nedd4 KO CrispR cell lines, analysis of knock-down cells and conditional mouse knockouts.

6. The authors discovered a bimodal distribution of ubiquitinated peptides in the absence of Nedd4. Reasons causing this bimodal distribution should be provided. For general readers, it is hard to believe that reduction of a single E3 ligase can cause such a significant effect on protein ubiquitination.

Response: We agree that this is very interesting. Indeed, there are very few papers that use a global approach to measure ubiquitination changes following loss of any given ubiquitin ligase. As such, any reader with an interest in ubiquitination would find our results of great interest. However, we do reference and discuss a recent example of profound global ubiquitination changes following loss of an E3 ligase (Rsp5 – yeast Nedd4 homolog) in the paper. Specifically, Fang et al showed that loss of Rsp5 strongly abrogated global K48 ubiquitination that would normally occur following heat shock, without any overall change in total ubiquitin levels (Nat Cell Biol, 2014 DOI: 10.1038/ncb3054). This effect appears to be very similar to what we see both in terms of reductions in K48 and K63 linkage usage and the corresponding bimodal distribution of observed ubiquitin remnant containing peptides. Furthermore, we discuss the possibility that the bimodal distribution may arise from Nedd4 affecting the ubiquitinome machinery in broader contexts within neural crest cells. Due to the large effect on ubiquitin usage following Nedd4 KD we independently performed proteomics analysis of total protein and ubiquitin remnant enrichment with an alternative siRNA. This analysis replicated the bimodal distribution of ubiquitinated peptides in Nedd4 KD cells, confirming this effect to be true. However, as this second experiment with an alternative siRNA had much lower coverage than our previous proteomics analyses it has not

been included in our current manuscript. We are happy to share this data with the reviewers if they wish to see it.

7. In Figure 5A, how is the number of the protein groups obtained? In Figure 5B, based on the scatter plot, the 3-fold increased and 3-fold reduced points do not follow the criteria described in the main text.

Response: This has now been moved to Figure 3. Each highlighted spot has 3-fold increased or 3-fold reduced ubiquitin usage.

8. The authors mentioned that their experiments provided a foundation for studies examining direct versus indirect Nedd4 targets, disentangling mono-ubiquitinated targets from poly-ubiquitinated, and determining exactly which poly-ubiquitin linkages are directed by Nedd4 on any given target. Detailed experiments should be provided to demonstrate this principle.

Response: This is no longer applicable.

9. In the motif search analysis, the authors searched the known motifs, PPXY and PY-like. Are there any new motifs in the differentially regulated proteins? How do these motifs affect the ubiquitination events regulated by Nedd4?

Response: We did not observe any significant enrichment of reported Nedd4 motifs using a variety of freely available motif searching tools.

10. The labels for the ratio in Figure 2D and 2E need to be corrected. The criteria for the change in the abundance at the protein level should be explicitly stated to show the differentially regulated proteins in Figure 2E.

Response: This is no longer applicable.

11. In the second panel of Supplementary Figure 2, both the non-modified and the ubiquitinated Erbb3 are increased after knocking down Nedd4. The authors should provide an explanation for this. In addition, the ratio for WB should also be provided.

Response: This analysis is of total protein after Nedd4 KD. Ratios for WB analysis is now provided alongside each antibody.

REVIEWER COMMENTS

Reviewer #2 (Remarks to the Author):

The authors have made a sincere and reasonable efforts to address the points raised. While the overall depth of the proteome and ubiquitylation profiling remains well below current state-of-the-art in the field. Nonetheless, the authors now focus on showing NEDD4-dependent ubiquitylation of specific substrates and the manuscript has an increased focus on biological roles of NEDD4-dependent ubiquitylation. In light of the change of the focus, and more stronger data on the validation of ubiquitylated proteins, the manuscript can be considered for publication in Nature Communications.

Reviewer #3 (Remarks to the Author):

This manuscript used quantitative analysis of the proteome, transcriptome and ubiquitinome to identify proteins, mRNA, and ubiquitinated proteins that are regulated by knocking down an E3 ligase NEDD4. The authors then identified key signaling pathways that NEDD4 regulates. They found novel NEDD4 targets and specific ubiquitin lysine linkages regulated by NEDD4. Through functional study, they demonstrated that NEDD4 mediates the ubiquitination of profilin1 and then regulates filamentous actin polymerization. Overall all the results seem convincing. However, many questions are not answered and many problems are present in the data.

1. The authors mentioned that mice lacking NEDD4 in NCCs have aberrant NCC death, decreased osteoblast proliferation and reduction in craniofacial bone and peripheral neurons. Are the identified proteins or ubiquitinated proteins closed associated with these functions?
2. The authors discovered that Pfn1 knockdown could reverse the effect of NEDD4 in actin remodeling. Is this effect caused by the ubiquitination of Pfn1 mediated by NEDD4? Functional studies of the ubiquitination resistance Pfn1 mutant is required to validate the role of Pfn1 ubiquitination.
3. The authors discovered that NEDD4 reduces the K63, K48 and K33 ubiquitin linkages. What are the roles of these regulated ubiquitin linkages in mediating the functions of NEDD4? In addition, regulation of K33 chain linkage by NEDD4 is not biochemically validated.

4. Why is there a predominant band below 37 kDa in Fig. 4B (blotted with HA for Ub)?

5. Figure 1: D: Please color protein that are up and down regulated. E: Are "reduced" and "increased" opposite?

6. Figure 2: C: Why are there two peaks? It is better to set the label for the x axis to integer numbers. E: Add standard deviation to the bar graph. F: Add a short line to indicate the exact position of molecular weight marker for the WB and Coomassie stain images.

7. Figure 3: A: K-epsilon-GG (not K-e-GG). The "total protein data" is misleading. Why are the down, unchanged, and up regulated proteins different from total protein numbers in the table?

8. Figure 4: A-B: WB: Add IPed GFP WB image. Current result does not know whether the IPed proteins have similar levels in different samples. In addition, the Input sample lacks the loading control. The signal for Ub-HA is too weak to determine whether the signal is changed or not. C: Lack of negative control for IP. Therefore, it cannot be determined whether the interaction is specific or not. E: Is Ub-Pfn1 changed or not? It is hard to tell from the image. Is the signal below 37 kDa seems GFP-Pfn1? Why is it also shown in the Ub blotting image? F: right panel: lack control; without statistic data. G: for quantification, the experiments should be done on the same blot. H: use h instead of hr.

9. Figure 6: D: add data points to the bar graph. Add more scales at y axis.

10. Figure 7: D: WB for WT and Nedd4^{-/-} should be done on the same blot for comparison.

11. Important proteins and ubiquitinated proteins should be labeled in proteome, ubiquitinome, and transcriptome data.

12. Citing a reference or a figure for the following sentence in page 7. Enrichment of genes that regulate the actin cytoskeleton is in strong agreement with the cell morphology change induced by Nedd4 knockdown.

13. Molecular weight marker should be provided, especially for the long images.

14. Missing loading control for Supplementary Fig. 5.

15. Supplementary Fig. 6: Why are all the points shifted to the left side of the image? Is normalization performed?

16. Proteomics data should be deposited to public database available for review.

17. There are some language problems. For example:

a) Quantitative data was obtained

b) quantitative data was available

c) Similar disruption to K48 linkages were observed

d) keep the units (hr, hrs, minutes) consistent with the standard usage

e) a notable correlate to our finding that Nedd4 regulates Pfn1 stability in NCCs 438 are the contrasting roles

f) 100ug/mL

g) in 1.5mL IAP. 1450μL of each peptide fractions was

h) Figure 5B, Figure 6D, Figure 7C: Fluorescence instead of Fluorosecence.

i) vi siRNA should be explained.

j) Use standard ways to write gene and protein names.

Reviewer #4 (Remarks to the Author):

Based on previous findings that ubiquitin ligase Nedd4 regulates neural crest development, this article seeks to understand the function of Nedd4 in neural crest cells. First, transcriptome, proteome, and ubiquitinome analysis were combined to identify Nedd4 targets in neural crest cells.

This combinatory approach is informative and 276 novel targets of Nedd4 in neural crest cells were identified. The majority of the ubiquitin targets are not regulated at protein abundance, leaving an interesting area for exploration. Next, the authors focused on one group of Nedd4 targets, actin cytoskeletal regulators and demonstrated that Nedd4 regulates Profilin1 turnover to control the assembly of actin stress fibers and neural crest cell migration. In general, while the article does not provide mechanistic insights of how Nedd4 regulates its target proteins, it reported novel ubiquitin targets for further analysis and a novel activity of Nedd4 in regulating actin dynamics. I think it will be of interest to a good range of readers. Therefore, I support its publication in Nature Communications after the following points are addressed.

1. Figure 1E: panel is mislabeled: on the left shows proteins significantly increased and on the right shows proteins significantly reduced.
2. Figure 1F: Only pathways enriched in genes with transcriptional changes are shown. How about pathways enriched in differentially expressed proteins? Is there overlaps in the enriched pathways?
3. Ln 185: it should be stated clearly that the 19 proteins showed over 3-fold changes in ubiquitination levels, but not protein abundance.
4. Figure 4B: Comparing Nedd4 transfected cells with Nedd4-CS transfected cells, while there is a dramatic increase in ubiquitinated Pfn1, there is also an evident increase in the level of Nedd4 and Ubiquitin. This should either be replaced or quantified. Similarly, the intensity of the ubiquitinated Pfn1 in Figure 4E should be quantified.
5. Is NCC migration reduced in neural tube culture (Fig. 6A)? Is the neural tube explant isolated from cranial level or trunk level? I think this is a better assay for NC migration comparing to transwell assay (Fig. 6B). Reduced number of cells that pass through the membrane which may actually reflect cells' response to chemoattractant, or ability to change shapes (or rigidity).
6. Fig 6A: Is the reduction of Sox10 expression quantified (mentioned in figure legend but not in text)? Is this a direct effect of Nedd4 KD or a secondary effect of impaired neural crest health?
7. Is neural crest migration at different axial levels differentially affected by Nedd4 KD? Why?

8. Figure 7 A-B: It seems that Nedd4 KO does not affect the recovery speed of actin filaments but rather change the density /amount of F-actin. Just as shown in Fig 5, at 60min, both control and Nedd4^{-/-} cells, there is already some short F-actin assembled. F-actin assembly seems completely recovered for both cells at 180min (better demonstrated by Fig. 7B). In combination with the result that siRNA for Pfn1 did reduced the F-actin of Nedd4^{-/-} cells to similar level as in control cells, and that Nedd4 regulate the level of Pfn1 protein (Fig6), a more appropriate conclusion may be that Nedd4 regulate the density of actin filaments through regulating the level of Pfn1.

9. Figure 7F: The increase of F/G actin ratio in Nedd4 KO mainly results from a dramatic reduction of G-actin, but there is not a significant increase of F-actin. Total actin should be examined to compare.

10. A reduction in K63, K48, and K33 ubiquitin linkages were observed upon Nedd4 KD, but only K63 and K48 linkages were described in the introduction. A description of K33 linkage should be included. Since K63 linkage often associates with non-proteasomal outcomes, this may partially explain the result that the abundance of most Nedd4 target proteins are not changed by Nedd4 KD. This should be elaborated in the result or discussion session with suggestions of how Nedd4 may regulate the activity of its target proteins.

Nature Communications comments 2021

We thank the reviewers for their thoughtful and constructive comments. We have now addressed all of their concerns as detailed below and in our revised manuscript (Reviewers' comments in black and our responses in blue). All alterations to the manuscript are highlighted in green.

Reviewer #3 (Remarks to the Author):

This manuscript used quantitative analysis of the proteome, transcriptome and ubiquitinome to identify proteins, mRNA, and ubiquitinated proteins that are regulated by knocking down an E3 ligase NEDD4. The authors then identified key signaling pathways that NEDD4 regulates. They found novel NEDD4 targets and specific ubiquitin lysine linkages regulated by NEDD4. Through functional study, they demonstrated that NEDD4 mediates the ubiquitination of profilin1 and then regulates filamentous actin polymerization. Overall all the results seem convincing. However, many questions are not answered and many problems are present in the data.

1. The authors mentioned that mice lacking NEDD4 in NCCs have aberrant NCC death, decreased osteoblast proliferation and reduction in craniofacial bone and peripheral neurons. Are the identified proteins or ubiquitinated proteins closely associated with these functions?

To address this question we now include process network analysis of Nedd4 ubiquitinated targets (Supplemental Table 5; Page 9, line 14-17) which demonstrates enrichment of several molecular pathways that are known to regulate NCC survival and craniofacial development. In addition to this analysis providing further support to a primary role for Nedd4 in regulating the actin cytoskeleton, it also identifies critical roles for Nedd4 in RNA transport and ribosome biogenesis, which when dysregulated lead to aberrant NCC death and well-known craniofacial defects such as Treacher Collins Syndrome and Diamond Blackfan anemia (Ross & Zerbali, 2014). We now expand our discussion to describe possible molecular pathways that underpin the phenotype of *Nedd4*^{-/-} embryos (Page 22, line 17-24 and Page 23, line 1-18).

2. The authors discovered that Pfn1 knockdown could reverse the effect of NEDD4 in actin remodeling. Is this effect caused by the ubiquitination of Pfn1 mediated by NEDD4? Functional studies of the ubiquitination resistance Pfn1 mutant is required to validate the role of Pfn1 ubiquitination.

This is a good suggestion. We now provide functional studies of ubiquitination resistant PFN1 mutants (Page 15, line 4-21). To identify PFN1 lysine residues ubiquitinated by NEDD4 we performed a cell-based ubiquitination assay with several lysine-to-alanine mutants and compared these to GFP-PFN1 (Fig. 8A-B). While our analysis identified reduced ubiquitination of GFP-PFN1-K108A, we also identified increased ubiquitination of GFP-PFN1-K127A. Such an increase is consistent with the K127A substitution destabilising protein structure, as observed for several other mutant versions of this protein (Wu et al., 2012). Indeed, K127 is located within the well conserved C-terminal alpha helix and this lysine residue forms several hydrogen bonds with a neighbouring alpha helix and loop region, which when replaced with an uncharged amino acid would likely alter protein integrity.

Decreased ubiquitination of GFP-PFN1-K108A was primarily restricted to the monoubiquitinated band at 50-52 kDa. To assess if ubiquitination of this lysine residue regulates protein function we performed cycloheximide chase assays and cytochalasin D actin repolymerisation assays with the mutant proteins (Fig. 8C-D). In comparison to GFP-PFN1 we found that GFP-PFN1-K70A and GFP-PFN1-K108A are partially stabilised in NCCs, therefore suggesting that these lysine residues participate in protein turnover. Moreover, we found that actin repolymerisation was significantly increased in NCCs expressing GFP-PFN1-K108A, similar to that observed in *Nedd4*^{-/-} NCCs. However, such an effect was not observed for the GFP-PFN1-K70A mutant, indicating that PFN1 functionality is regulated by ubiquitination of specific lysine residues. A surprising outcome of our new work is that overexpression of GFP-PFN1 had only a minor and non-significant increase in actin repolymerisation when compared to un-transfected control NCCs. Taken in light of our result that *Pfn1* knockdown normalises actin repolymerisation in *Nedd4*^{-/-} NCCs, this suggests that site-selective NEDD4-mediated ubiquitination of PFN1 regulates both functional outcomes as well as protein abundance, and that both play critical roles in actin polymerisation. This is now discussed in our revised manuscript (Page 21, line 10-24 and Page 22, line 1-9).

3. The authors discovered that NEDD4 reduces the K63, K48 and K33 ubiquitin linkages. What are the roles of these regulated ubiquitin linkages in mediating the functions of NEDD4? In addition, regulation of K33 chain linkage by NEDD4 is not biochemically validated.

Upon further analysis of K33 linkages in our MS data we found that these events were only detected in 1 repeat. For this reason we have now removed K33 linkage from our analysis as this is not quantitative. NEDD4 has been considered to preferentially ubiquitinate substrate proteins through K63 conjugation, thus, a reduction in this linkage type after *Nedd4* knockdown in NCCs was expected. Moreover, recent studies of single proteins (Wan et al., 2021) and global ubiquitination (Fang et al., 2014) have highlighted that NEDD4 also mediates K48 linkages, which is also consistent with the reduction of this linkage type after *Nedd4* siRNA knockdown. In general, K48-linked polyubiquitin chains promote degradation of target proteins through the proteasome, whereas K63-linkages are associated with proteasome independent processes, such as endocytosis, inflammatory signalling, DNA repair and autophagy. K63 ubiquitin chains can also assist the assembly of K48/K63 branched chains, thus target degradation by the proteasome (Ohtake et al., 2018). Our matched analysis of ubiquitination sites with total protein abundance identified several putative NEDD4 substrates with a corresponding and significant increase in protein level which provides confidence in our finding that both K48 linkages and K63 linkages are affected. While only a small percentage of putative targets had increased protein abundance, it is likely that many of the NEDD4 mediated K48 linkages are context dependent and that altered protein stability initiates feedback mechanisms to maintain cell homeostasis. Indeed, strong support of NEDD4 directing K48 linkages comes from our directed analysis of PFN1 in NCCs. Although PFN1 did not have increased abundance in our total proteome data, our cell culture analysis found that it was stabilised in NCCs lacking *Nedd4* and that modest increases in protein levels were coincident with reduced RNA expression. Our addition of new data from ubiquitination-resistant PFN1 mutants also suggests that some NEDD4-directed ubiquitination events are functional in nature rather than degradative. While further analysis of additional targets identified in our study is

required to understand the full extent of NEDD4 directed K48 and K63 linkages, we are confident that our analysis provides significant new insight to the role of NEDD4 in regulating the NCC ubiquitinome. We now expand our discussion of the possible roles of K48 and K63 linkages in the revised manuscript (Page 16, line 18-22 and Page 17, line 1-21).

4. Why is there a predominant band below 37 kDa in Fig. 4B (blotted with HA for Ub)?

Upon re-examining our raw data we realised that the molecular weight markers were incorrectly labelled in Fig. 4B which has now been corrected in the revised manuscript. To answer the reviewers question we provide an annotated immunoblot (Reviewer Fig. 1, below) showing the raw molecular weight markers next to GFP-PFN1 input immunoblotted for GFP, and GFP IP immunoblotted for HA (to detect ubiquitin-HA specific to PFN1). These immunoblots identify a major GFP-PFN1 band at around 40-42 kDa and a predominant ubiquitinated GFP-PFN1 band at 50-52 kDa. Given the size increase of 8-10 kDa, which matches the molecular weight of ubiquitin, this strongly suggests the predominant band at 50-52 kDa represents GFP-PFN1 that is monoubiquitinated by NEDD4. This is now detailed in the revised manuscript (Page 11, line 8-10). To make this clearer in the main figures we have modified Fig. 4A (which replaces the original Fig. 4B) to show extended western blots of HA-ubiquitin and include multiple molecular weight markers.

5. Figure 1: D: Please color protein that are up and down regulated. E: Are "reduced" and "increased" opposite?

To clarify our point that very few proteins have matching mRNA expression profiles we now include a new Fig. 1E that only represents proteins that are significantly up- or down-regulated after *Nedd4* siRNA knockdown, and colour code matched mRNAs that are also up or down regulated. The reviewer is correct that the original files incorrectly labelled the reduced and increased proteins. We hope this new figure makes it easier to understand the major point of the data.

6. Figure 2: C: Why are there two peaks? It is better to set the label for the x axis to integer numbers.

This is a good suggestion. We now relabel the x axis in Fig. 2C to include integer numbers. Our interpretation of the data is that there are two peaks because of a global shift in ubiquitin linkages arising after removal of *Nedd4*. The ubiquitinated sites that are decreased are likely to represent both direct and indirect targets of *Nedd4* as our data also uncovered several other E3 ligases with deficient ubiquitination that may alter their capacity to ubiquitinate their own substrates. We propose that the global shift in ubiquitinated targets also arises due to a predominant role of *Nedd4* in NCCs, thus removal of this E3 ligase leads to a profound effect on ubiquitination of a large number of proteins. While we had initially thought that a global shift in ubiquitinated peptides was unexpected, there are several examples of similar global shifts in H/L ratios from ubiquitin enrichment studies performed on knockdown cells (Lee et al., 2011; Tong et al., 2014). We now expand our discussion to explore the reasons for two peaks in the data (Page 16, line 18-22, Page 17, line 1-21 and Page 18, line 11-24 and Page 19, line 1-3).

E: Figure 2: E: Add standard deviation to the bar graph.

Standard deviation is now included in Fig. 2E which was calculated using H/L ratios for each peptide across our MS replicates. In deriving this standard deviation we found that K33

linkages were only detected in one MS replicate and for this reason have now removed this linkage type from our analysis.

F: Add a short line to indicate the exact position of molecular weight marker for the WB and Coomassie stain images.

This has now been corrected.

7. Figure 3: A: K-epsilon-GG (not K-e-GG). The "total protein data" is misleading. Why are the down, unchanged, and up regulated proteins different from total protein numbers in the table?

We apologise for the disparity between the table and the text. This table has now been replaced with a new version to make the breakdown of peptides and proteins clearer. In addition, we have reworded the results section to match the number of proteins that are significantly up- or down-regulated, as presented in full in the complete dataset (Supplemental table 1 and 6; Page 9, line 8-10).

8. Figure 4: A-B: WB: Add IPed GFP WB image. Current result does not know whether the IPed proteins have similar levels in different samples. In addition, the Input sample lacks the loading control. The signal for Ub-HA is too weak to determine whether the signal is changed or not. C: Lack of negative control for IP. Therefore, it cannot be determined whether the interaction is specific or not. E: Is Ub-Pfn1 changed or not? It is hard to tell from the image. Is the signal below 37 kDa seems GFP-Pfn1? Why is it also shown in the Ub blotting image? F: right panel: lack control; without statistic data. G: for quantification, the experiments should be done on the same blot. H: use h instead of hr.

We thank the reviewer for these suggestions and have made all of the changes in revised Fig. 4. To align this figure with the main message of the manuscript we have moved the ubiquitination analysis of KIF16B to Supplemental Fig. 6.

Fig.4A now includes ubiquitination analysis of PFN1 with appropriate loading controls and quantitation from 4 separate experiments, which shows significant increase in PFN1 ubiquitination in the presence of NEDD4 compared to NEDD4-CS.

Fig. 4B replaces Fig. 4C and now includes a new experiment confirming the interaction between NEDD4 and PFN1. This experiment includes NEDD4 and NEDD4CS, and several negative controls including different combinations of expression constructs and negative control immunoprecipitation with mouse IgG.

Fig. 4D now replaces Fig. 4E. To confirm that PFN1 is ubiquitinated by NEDD4 we attempted to immunoprecipitate PFN1 from WT and *Nedd4*^{-/-} NCCs. However, in our hands we failed to immunoprecipitate a sufficient quantity of PFN1 with this approach. To overcome this limitation we transfected GFP-Pfn1 into WT and *Nedd4*^{-/-} NCCs, isolated GFP-PFN1 with GFP beads, and immunoblotted for total ubiquitin. In the original image there was a major band at around 37 kDa which we agree is unexpected, but does occasionally appear in immunoblots after IP of GFP-PFN1 with GFP-beads. We have now repeated this experiment several times without addition of MG132 prior to cell lysis as we found that inhibition of the proteasome pushes GFP-PFN1 into an insoluble fraction, consistent with the findings of Smith et al. (2015). Our new results lack a predominant band at 37 kDa, with the majority of ubiquitination seen above 50 kDa, consistent with our assays in Fig. 4A. Quantitation of the ubiquitinated products from 3X WT and 3X *Nedd4*^{-/-} NCC lines identifies a significant decrease in PFN1 ubiquitination in the absence of NEDD4.

The abundance of PFN1 in WT and Nedd4^{-/-} NCCs is now included in Fig. 4C, with quantitation from 3X WT and 3X Nedd4^{-/-} NCC lines. Original Fig. 4G is now labelled Fig. 4E. For quantitation of PFN1 protein levels after CHX chase (Fig. 4F) we performed the experiments on all NCC lines and ran them on the same blot, which is included as Supplemental Fig. 6B. hr has been replaced with h in all new figures.

9. Figure 6: D: add data points to the bar graph. Add more scales at y axis. We have now expanded the number of repeats in this experiment and include data points on the graph.

10. Figure 7: D: WB for WT and Nedd4^{-/-} should be done on the same blot for comparison. We now replace this figure with the immunoblot on the same membrane.

11. Important proteins and ubiquitinated proteins should be labeled in proteome, ubiquitinome, and transcriptome data. KIF16B and PFN1 are now identified in Fig. 3B.

12. Citing a reference or a figure for the following sentence in page 7. Enrichment of genes that regulate the actin cytoskeleton is in strong agreement with the cell morphology change induced by Nedd4 knockdown.

We have now changed the wording of this sentence to read “... alterations to genes that regulate the actin cytoskeleton is in strong agreement with the cell morphology change induced by Nedd4 knockdown” and include an appropriate reference (Pollard and Cooper, 2009) (Page 7, line 20-21).

13. Molecular weight marker should be provided, especially for the long images. This is a good suggestion. We now provide multiple molecular weight markers on all long images.

14. Missing loading control for Supplementary Fig. 5. We now include a loading control for the total Ub immunoblot.

15. Supplementary Fig. 6: Why are all the points shifted to the left side of the image? Is normalization performed?

Supplementary Fig. 6 is replaced by Supplementary Fig. 5 in the revised manuscript. This experiment was performed with NEDD4 immunoprecipitation from LIGHT SILAC labelled NCCs and compared to an IgG control immunoprecipitation from HEAVY SILAC NCCs. The data are presented without Maxquant normalisation of H/L ratios as we expected any NEDD4 interacting proteins to be of low abundance compared to IgG peptides from the antibody. Thus, normalisation within Maxquant would favour any differences in the IgG peptides when trying to centre the H/L ratio on zero for normal distribution of the data which would affect our ability to identify interacting proteins. As global normalisation was not performed within Maxquant we expect any NEDD4 interacting protein to be on the left-hand side of the graph. In analysing these data we considered anything with a greater than 1.5-fold enrichment compared to IgG control to be an interactor of NEDD4 (Supplementary Table 7 and 8). While modest, we based a 1.5-fold threshold on this dataset because of the

presence of known Nedd4 interacting proteins at or below this threshold (eg AP-1 complex subunit μ). We have now replaced Supplemental Table 8 to clarify the proteins that were identified as Nedd4 interactors in this analysis.

16. Proteomics data should be deposited to public database available for review.

The mass spectrometry proteomics data have been deposited to the ProteomeXchange with identifier PXD024103. Reviewer account details:

Accession number: PXD024103

Username: reviewer_pxd024103@ebi.ac.uk

Password: PbqJgrud

This is now included in Methods section.

17. There are some language problems. For example:

a) Quantitative data was obtained. This has been amended throughout the text.

b) quantitative data was available. This has been amended.

c) Similar disruption to K48 linkages were observed. This has been removed.

d) keep the units (hr, hrs, minutes) consistent with the standard usage. This has been amended throughout the text.

e) a notable correlate to our finding that Nedd4 regulates Pfn1 stability in NCCs 438 are the contrasting roles. This has been reworded.

f) 100ug/mL. All concentrations have been amended.

g) in 1.5mL IAP. 1450 μ L of each peptide fractions was. This has been amended.

h) Figure 5B, Figure 6D, Figure 7C: Fluorescence instead of Fluorosecence. This has been amended.

i) vi siRNA should be explained. This has been amended.

j) Use standard ways to write gene and protein names. All gene and protein names have been corrected.

Reviewer #4 (Remarks to the Author):

Based on previous findings that ubiquitin ligase Nedd4 regulates neural crest development, this article seeks to understand the function of Nedd4 in neural crest cells. First, transcriptome, proteome, and ubiquitinome analysis were combined to identify Nedd4 targets in neural crest cells. This combinatory approach is informative and 276 novel targets of Nedd4 in neural crest cells were identified. The majority of the ubiquitin targets are not regulated at protein abundance, leaving an interesting area for exploration. Next, the authors focused on one group of Nedd4 targets, actin cytoskeletal regulators and demonstrated that Nedd4 regulates Profilin1 turnover to control the assembly of actin stress fibers and neural crest cell migration. In general, while the article does not provide mechanistic insights of how Nedd4 regulates its target proteins, it reported novel ubiquitin targets for further analysis and a novel activity of Nedd4 in regulating actin dynamics. I think it will be of interest to a good range of readers. Therefore, I support its publication in Nature Communications after the following points are addressed.

1. Figure 1E: panel is mislabeled: on the left shows proteins significantly increased and on the right shows proteins significantly reduced.

This has now been replaced with a new figure to help clarify our point that there is limited correlation between protein and RNA abundance.

2. Figure 1F: Only pathways enriched in genes with transcriptional changes are shown. How about pathways enriched in differentially expressed proteins? Is there overlaps in the enriched pathways?

This is a good suggestion. We now provide process network analysis of: 1) proteins with altered abundance, and 2) NEDD4 targets (Supplemental Table 4 and 5). While there is little overlap between the differentially expressed genes and proteins, there is a high degree of overlap between the transcriptional changes and NEDD4 ubiquitinated targets. Notably, this includes several signalling pathways and regulation of the actin cytoskeleton, which helps to strengthen our analysis. This is now discussed in the relevant sections of the revised manuscript (Page 7, line 17-20 and Page 9, line 14-17).

3. Ln 185: it should be stated clearly that the 19 proteins showed over 3-fold changes in ubiquitination levels, but not protein abundance.

This has now been reworded (Page 9, line 17-19).

4. Figure 4B: Comparing Nedd4 transfected cells with Nedd4-CS transfected cells, while there is a dramatic increase in ubiquitinated Pfn1, there is also an evident increase in the level of Nedd4 and Ubiquitin. This should either be replaced or quantified. Similarly, the intensity of the ubiquitinated Pfn1 in Figure 4E should be quantified.

This is a good suggestion. We now provide quantification of the ubiquitination induced by overexpression of NEDD4 compared to NEDD4-CS from 4 separate experiments in Fig. 4A. We also provide quantification of GFP-PFN1 ubiquitination from all *Nedd4*^{-/-} NCC lines in Fig. 4D.

5. Is NCC migration reduced in neural tube culture (Fig. 6A)? Is the neural tube explant isolated from cranial level or trunk level? I think this is a better assay for NC migration comparing to transwell assay (Fig. 6B). Reduced number of cells that pass through the membrane which may actually reflect cells' response to chemoattractant, or ability to change shapes (or rigidity).

This is a good suggestion. Neural tube explants were taken from the trunk level (now clarified in the methods and text; Page 12, line 20 and Page 26, line 1). We performed real time tracking of NCCs from trunk neural tube explants treated with control or *Nedd4* siRNA to address this question. Our new results show that the migration distance of NCCs lacking *Nedd4* is reduced compared to controls. We have now replaced Fig. 6B with these data and moved the transwell migration assay with *Nedd4*^{-/-} NCCs to Supplemental Fig. 7 (Page 13, line 1-2).

6. Fig 6A: Is the reduction of Sox10 expression quantified (mentioned in figure legend but not in text)? Is this a direct effect of *Nedd4* KD or a secondary effect of impaired neural crest health?

In our previous analysis of *Nedd4*-deficient NCCs (Wiszniak et al., Dev Biol, 2013) we carefully analysed Sox10 expression by qRT-PCR, immunoblotting and immunostaining with cell lines, primary NCC explants and mouse embryos. Our results showed that Sox10 expression is reduced at both the RNA and protein level and that this expression was unable to be normalised with the addition of the cell death inhibitor Z-VAD-FMK. We also found that SOX10 expression is reduced prior to cell death, which suggests that the loss of Sox10 is a direct effect of *Nedd4* KD, rather than impaired health of the cells. Within our global datasets of *Nedd4* KD NCCs we also identify reduced expression of Sox10 mRNA, reduced levels of SOX10 protein, and putative NEDD4 ubiquitination sites on SOX10. While we have not explored this as a direct NEDD4 ubiquitinated target, our data could suggest that NEDD4 regulates the function of SOX10. However, at this stage we prefer not to extrapolate on these preliminary observations without further validation and functional investigation.

7. Is neural crest migration at different axial levels differentially affected by *Nedd4* KD? Why?

In our current analysis we have carefully quantified trunk NCC migration at each somite level of E9.5 embryos to capture different stages of NCC development. Within the trunk, the timing of NCC delamination and migration away from the neural tube is tightly correlated with somitogenesis, such that "younger" somites contain fewer NCCs that have migrated less than NCCs in "older" somites. After delamination the earliest population of NCCs migrate within the somites toward the dorsal aorta where they will stop and begin their differentiation into sympathetic neurons. Thus, "younger" somites contain NCCs that are actively migrating to the dorsal aorta while "older" somites contain NCCs that have migrated vast distances to reach the dorsal aorta where they condense into the sympathetic chain. Our data indicate that the distance migrated by trunk NCCs within the "younger" somites is reduced in the absence of *Nedd4* (i.e. NCCs within somites 1-9 that contain actively migrating NCCs that have not reached their target site of the dorsal aorta). The fact that we do not see major differences in the more mature "older" somites (i.e. somites 11-14) suggests that NCCs lacking *Nedd4* can still migrate to their respective target sites, but with altered migration dynamics. Thus, rather than showing that migration is affected

differently at different axial levels, we interpret our in vivo data to suggest that Nedd4 regulates F-actin abundance to regulate a large proportion of actively migrating NCCs. Indeed, our previous analysis of cranial NCCs (Wisznia et al., Dev Biol, 2013) in which we found reduced expression of NCC markers in the anterior regions of the head of E8.5 *Nedd4*^{-/-} embryos, is consistent with migratory deficiencies in cranial NCCs. Taken together this suggests that migration defects occur at many axial levels in *Nedd4*^{-/-} embryos. We now expand our discussion of migration defects in the revised manuscript to address this point (Page 20, line 17-24 and Page 21, line 1-3).

8. Figure 7 A-B: It seems that Nedd4 KO does not affect the recovery speed of actin filaments but rather change the density /amount of F-actin. Just as shown in Fig 5, at 60min, both control and Nedd4^{-/-} cells, there is already some short F-actin assembled. F-actin assembly seems completely recovered for both cells at 180min (better demonstrated by Fig. 7B). In combination with the result that siRNA for Pfn1 did reduce the F-actin of Nedd4^{-/-} cells to similar level as in control cells, and that Nedd4 regulates the level of Pfn1 protein (Fig6), a more appropriate conclusion may be that Nedd4 regulates the density of actin filaments through regulating the level of Pfn1.

This is a good suggestion. We have now reworded the text to indicate that NEDD4 regulates the density of actin filaments (Page 12, line 17 and Page 14, line 8-10).

9. Figure 7F: The increase of F/G actin ratio in Nedd4 KO mainly results from a dramatic reduction of G-actin, but there is not a significant increase of F-actin. Total actin should be examined to compare.

We agree with this observation. The original image included in the manuscript has now been replaced by another example that is more representative of our repeats which shows that the alteration to the F/G actin ratio is both a reduction in G-actin and an increase in F-actin, as seen in the steady state levels. For consistency between images we have also replaced Fig. 7E which has the same loading orientation of F- and G-actin. We now provide a graph of F+G-actin quantitation that shows there is no major difference in “total” actin between control and *Nedd4*^{-/-} cells (Page 15, line 3-4).

10. A reduction in K63, K48, and K33 ubiquitin linkages were observed upon Nedd4 KD, but only K63 and K48 linkages were described in the introduction. A description of K33 linkage should be included.

As we were unable to see K33 linkages in all of our proteomics analyses we have now removed K33 linkage from our manuscript.

11. Since K63 linkage often associates with non-proteasomal outcomes, this may partially explain the result that the abundance of most Nedd4 target proteins are not changed by Nedd4 KD. This should be elaborated in the result or discussion session with suggestions of how Nedd4 may regulate the activity of its target proteins.

We agree with this interpretation and now expand our discussion of the data to specifically address the potential roles that Nedd4 plays through K48 and K63 linkages (Page 16, line 18-22 and Page 17, line 1-21).

Reviewer Figure 1. Ubiquitination of PFN1. 293T cells were transfected with HA-Ub, GFP-Pfn1 and FLAG-Nedd4 or FLAG-Nedd4-CS. (Left) Raw GFP western blot of input with molecular weight markers identifies GFP-PFN1 at 40-42 kDa. (Right) Raw HA western blot of GFP immunoprecipitation with molecular weight markers identifies ubiquitinated forms of GFP-PFN1 above 50 kDa. The band at 50-52 kDa represents monoubiquitinated PFN1.

REVIEWERS' COMMENTS

Reviewer #3 (Remarks to the Author):

The revised manuscript has been significantly improved after revision and the revised manuscript is acceptable for publication.

Reviewer #4 (Remarks to the Author):

The authors have addressed most of my concerns, I do not have additional critiques regarding the article.